# Deep multiomics profiling of brain tumors identifies signaling networks downstream of cancer driver genes

Hong Wang [1,2,3], Alexander K. Diaz[3,4], Timothy I. Shaw[2,5], Yuxin Li[1,2,4], Mingming Niu[1,4], Ji-Hoon Cho[2], Barbara S. Paugh[4], Yang Zhang[6], Jeffrey Sifford[1,4], Bing Bai[1,4,10], Zhiping Wu[1,4], Haiyan Tan[2], Suiping Zhou[2], Laura D. Hover[4], Heather S. Tillman [7], Abbas Shirinifard[8], Suresh Thiagarajan[9], Andras Sablauer [8], Vishwajeeth Pagala[2], Anthony A. High[2], Xusheng Wang [2], Chunliang Li [6], Suzanne J. Baker[4] & Junmin Peng [1,2,4]

High throughput omics approaches provide an unprecedented opportunity for dissecting molecular mechanisms in cancer biology. Here we present deep profiling of whole proteome, phosphoproteome and transcriptome in two high-grade glioma (HGG) mouse models driven by mutated RTK oncogenes, *PDGFRA* and *NTRK1*, analyzing 13,860 proteins and 30,431 phosphosites by mass spectrometry. Systems biology approaches identify numerous master regulators, including 41 kinases and 23 transcription factors. Pathway activity computation and mouse survival indicate the *NTRK1* mutation induces a higher activation of AKT downstream targets including MYC and JUN, drives a positive feedback loop to up-regulate multiple other RTKs, and confers higher oncogenic potency than the *PDGFRA* mutation. A mini-gRNA library CRISPR-Cas9 validation screening shows 56% of tested master regulators are important for the viability of *NTRK*-driven HGG cells, including TFs (Myc and Jun) and metabolic kinases (AMPK*a*1 and AMPK*a*2), confirming the validity of the multiomics integrative approaches, and providing novel tumor vulnerabilities.

[1] Department of Structural Biology, St. Jude Children's Research Hospital, Memphis, TN 38105, USA. [2] Center for Proteomics and Metabolomics, St. Jude Children's Research Hospital, Memphis, TN 38105, USA. [3] Integrated Biomedical Sciences Program, University of Tennessee Health Science Center, Memphis, TN 38163, USA. [4] Department of Developmental Neurobiology, St. Jude Children's Research Hospital, Memphis, TN 38105, USA. [5] Department of Computational Biology, St. Jude Children's Research Hospital, Memphis, TN 38105, USA. [6] Department of Tumor Cell Biology, St. Jude Children's Research Hospital, Memphis, TN 38105, USA. [7] Department of Pathology, St. Jude Children's Research Hospital, Memphis, TN 38105, USA. [8] Department of Information Sciences, St. Jude Children's Research Hospital, Memphis, TN 38105, USA. [9] Department of Diagnostic Imaging, St. Jude Children's Research Hospital, Memphis, TN 38105, USA. [10]Present address: Department of Laboratory Medicine, Nanjing Drum Tower Hospital, Nanjing University Medical School, Nanjing, Jiangsu 210008, China. Correspondence and requests for materials should be addressed to S.J.B. (email: Suzanne.Baker@stjude.org) or to J.P. (email: Junmin.Peng@stjude.org)

A central gap in cancer biology concerns how oncogenes drive the rewiring of molecular signaling networks to execute phenotypic changes[1–3]. Initial attempts to decode the molecular networks were through proteomic characterization via antibody-based approaches (e.g. the reverse phase protein array). However, these targeted approaches are restricted by profiling breadth and depth, largely due to antibody availability and specificity. Because signaling networks are highly regulated by protein posttranslational modifications, and particularly phosphorylation, phosphoproteomic analyses are indispensable for studying cancer signaling[4]. Recently, mass spectrometry (MS)-based proteomics technology has been emerging as the mainstream strategy for unbiased analysis of the genome-wide proteome and phosphoproteome[5,6]. Together with advanced DNA sequencing, these methodologies provide an unprecedented opportunity for deep omics analysis. It is now possible to integrate transcriptome, deep proteome and phosphoproteome to dissect oncogenic signaling networks, broadening our understanding of cancer biology[1–3,7–10].

High-grade gliomas (HGG) are the most prevalent malignant brain tumors, and confer devastating mortality[11]. Although significant efforts in glioma sequencing have unveiled comprehensive genome-wide mutation landscapes[11–18], a complete understanding of how genomic alterations lead to dysregulation of particular master regulators and specific pathways remains unclear. Previous HGG proteomic and phosphoproteomic studies extend our understanding of HGG signaling[19,20], but most of these attempts have used proteomic approaches of relatively shallow depth. There is essentially no deep HGG proteomic landscape available for the cancer research community. Here, for the first time, we present a new paradigm of identifying ~12,000 gene products (proteins) and >30,000 phosphosites at a false discovery rate (FDR) of approximately 1% for dissecting HGG cancer biology.

In the present paper, we compared two HGG mouse models driven by oncogenic receptor tyrosine kinases (RTKs: *PDGFRA D842V* and *TPM3-NTRK1* fusion), using integrative systems biology analyses of proteome, phosphoproteome and transcriptome. Mutations and/or amplifications of platelet-derived growth factor receptor alpha (PDGFRA) and fusion genes of the neurotrophic tyrosine receptor kinase (NTRK) have often been identified in pediatric and adult HGG[11,13,17,21–23]. We used engineered mouse HGGs expressing the mutated RTKs to examine signaling networks downstream of these cancer driver genes. With an integrated bioinformatics pipeline, we identified various functional modules and master regulators that are rewired in HGGs and demonstrated that the *TPM3-NTRK1* oncogene upregulates multiple other RTKs to form a positive feedback loop within the PI3K-AKT pathway, driving more rapid tumor development compared with the *PDGFRA*-driven HGG. Finally, we performed CRISPR-Cas9 validation screening to show that multiple master regulators that are RTK-PI3K-AKT downstream transcription factors (TF) and key metabolic sensor kinases are crucial for the survival of the *NTRK*-driven HGG cells.

## Results

### Deep proteome and phosphoproteome profiling of mouse HGGs

To generate a deep mouse HGG proteome and phosphoproteome landscape, we used our newly developed MS pipeline with extensive peptide separation power and high mass resolution[24–27]. Mouse HGG samples were generated by intracranial implantation of p53-null primary astrocytes transduced with either *PDGFRA D842V* mutation or the *TPM3-NTRK1* fusion, two oncogenic RTKs found in human HGGs (Fig. 1a, referred to as *PDGFRA* HGG and *NTRK* HGG, respectively). Both models

generated HGGs with highly mitotic pleomorphic tumor cells, many with features of astrocytic differentiation. The HGGs grew as focal masses with clear areas of invasion into the surrounding parenchyma at the boundaries of the tumor (Fig. 1b)[13,22]. The HGG and normal mouse cortex (control) samples were submitted to proteome, phosphoproteome and transcriptome profiling. Tandem mass tag (TMT) labeling was used to enable massively parallel proteome and phosphoproteome quantification of ten samples (Fig. 1c). Extensive basic pH reverse phase liquid chromatography (LC) prefractionation followed by an ultra-long acidic pH reverse phase LC were applied to facilitate maximal peptide separation. As a result, 13,860 proteins (11,941 gene products, 200,454 peptides and 3,264,804 MS2 scans) and 30,431 phosphosites (5959 phosphoproteins, 45,574 phosphopeptides, 1,829,889 MS2 scans) were identified (<1% FDR, Fig. 1c). Among them, 13,567 proteins (11,718 gene products) and 28,527 phosphosites were quantified in every sample (Supplementary Data 1 and 2), representing one of the deepest HGG proteomic datasets available to date.

To evaluate the quality of the datasets, we examined the MS-based results of the two transduced human oncogenes compared with phosphorylation events assayed by Western blotting as reported in our previous study[22], as well as the classification of all measurements. The protein expression levels of exogenous human *PDGFRA D842V* and *TPM3-NTRK1* agreed with the HGG genotypes (Fig. 1d). MS data of specific phosphosites were also consistent with immunoblot assays described previously in these HGG mouse models: AKT S473, PRAS40 T247, PDGFRA Y742, S6 S235 and S6 S236[22] (Supplementary Fig. 1). Principal component analyses and hierarchical clustering analyses revealed that the two RTK oncogenes drive distinct proteome, phosphoproteome and transcriptome profiles (Fig. 2a–d). In the MS analysis, the intragroup replicate samples showed minimal variations with low standard deviation, whereas the inter-group comparisons exhibited differences with a much larger standard deviation (Supplementary Fig. 2a, b). For transcriptome profiling, RNAseq replicates from a second cohort of HGGs displayed high reproducibility of these HGG mouse models ($R^2 > 0.95$, Supplementary Fig. 3a, b, Supplementary Data 3). Furthermore, the levels of expression of mutant PDGFRA or NTRK fusion genes expressed in the mouse HGGs are relevant to the levels of expression found in human tumor cells with these mutations (Supplementary Fig. 3c, d). Together, these results indicate the high quality of our omics datasets and demonstrate that the two oncogenic RTKs drive HGGs with reproducibly distinct global proteome and phosphoproteome profiles.

We further analyzed the correlation and profiling depth of proteome and transcriptome. The transcript levels and protein abundances showed moderate correlation (Fig. 2e, $R^2 = 0.5$), consistent with previously reported datasets[28,29]. We first applied a cutoff of FPKM > 1 for the transcriptome to filter out low-quality data. In 12,842 accepted transcripts, 10,838 (84%) corresponding proteins were mapped by MS (Supplementary Fig. 4a). Next, we investigated peptide coverage of each protein in this unbiased proteomics analysis. More than 96% of proteins were identified by at least two peptides (Supplementary Fig. 4b), and the average coverage of theoretically observable protein sequences reached 42% (Supplementary Fig. 4c). In addition, we estimated the phosphoproteomic profiling depth by comparing to all previous curated mouse phosphosites in the PhosphoSitePlus database, the most comprehensive protein modification database. Our phosphoproteome covered approximately 68% of the mouse phosphosites collected from all cell types and tissues, and contained 12,354 novel phosphosites not in the database. In summary, these data present a paradigm of one of the deepest proteome and phosphoproteome analyzed in cancer studies.

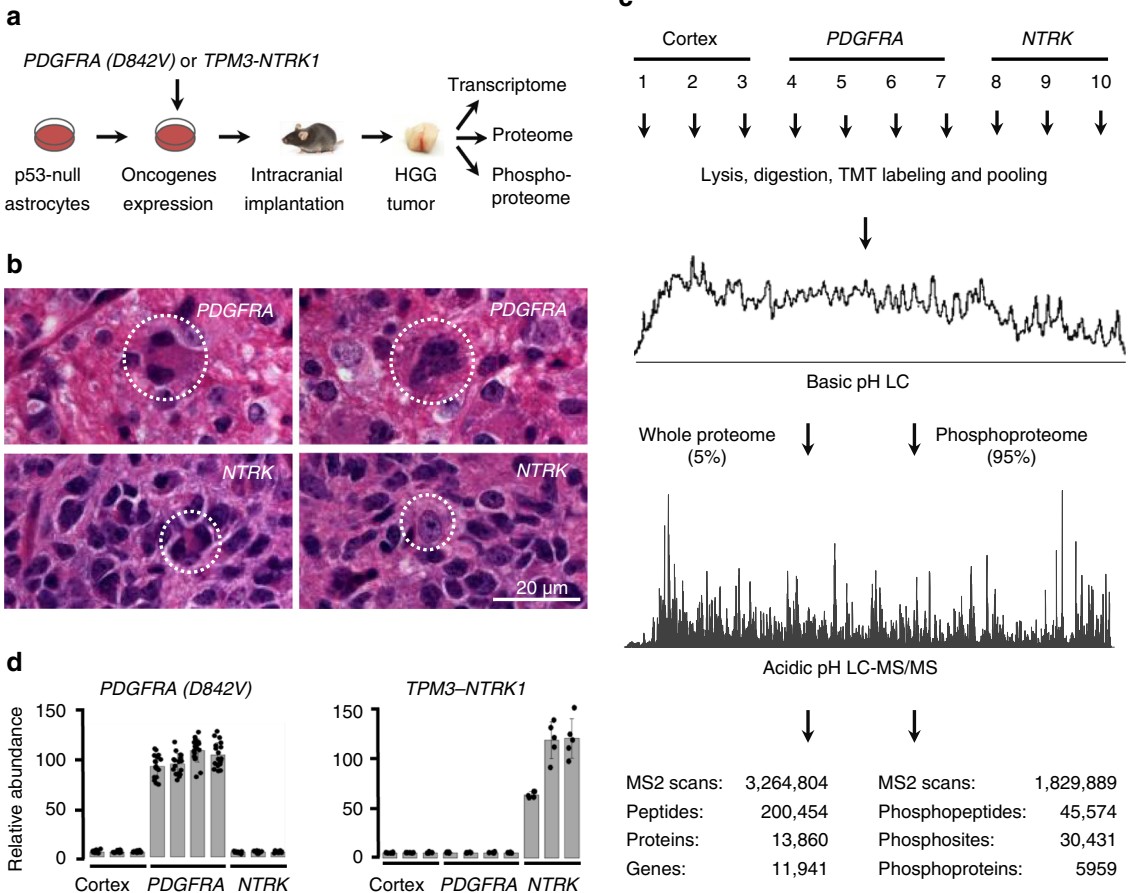

**Fig. 1** Deep quantitative omics analysis of HGG mouse models. **a** Overview of HGG mouse model analysis. Oncogene-transduced, p53-null primary mouse astrocytes were implanted into athymic nude mice to generate HGG tumors for analyzing proteome, phosphoproteome, and transcriptome. **b** High-grade features detected in the mouse gliomas. Hematoxylin and eosin images highlight *PDGFRA*-driven HGG with mitotic figures (left) and a multinucleated giant cell (right), as well as *NTRK*-driven HGG with mitotic figures (left) and tumor invasion of normal parenchyma, as evidenced by entrapped native neurons (right). A 20 μm scale bar is shown. **c** Proteomic workflow of 10-plex TMT-LC/LC-MS/MS. The same samples were used for profiling whole proteome and phosphoproteome. Three normal mouse cortex samples, four *PDGFRA* D842V driven HGG samples and three *TPM3-NTRK1* fusion driven HGG samples were dissected for the omics analyses. **d** MS validation of cancer driver gene expression. Protein expression of cancer drivers *PDGFRA* D8424V and *TPM3-NTRK1* were quantified by human-specific peptide sequences. Low levels of noise signal were observed due to co-isolated TMT-labeled species in the analysis. Error bar indicates s.d. HGG high-grade glioma, MS mass spectrometry, TMT tandem mass tag

**Proteomic analyses reveal HGG network modules and pathways.** We first identified differentially expressed (DE) proteins in mouse cortex, *PDGFR* and *NTRK* HGGs, and performed gene coexpression clustering, pathway analysis, and functional module classification by WGCNA[30] and ClueGO packages[31] (Fig. 3a). A total of 4703 DE proteins and 6768 DE phosphosites (2301 phosphoproteins) were identified and distributed into five whole proteome coexpression clusters (WP-C) and five phosphoproteome coexpression clusters (PP-C) respectively (Fig. 3b, c, Supplementary Data 1 and 2), leading to 67 functional modules (Supplementary Data 4a, b). As expected, the two largest modules rewired in tumors compared with normal cortex are cell cycle (in WP-C1, Fig. 3d) associated with tumor cell proliferation, and the PI3K signaling cascade (in PP-C2, Fig. 3e), which transduces signals downstream of RTKs. Collectively, a series of module groups including cancer signaling, gene expression, cell adhesion, metabolism, and neuronal functions are rewired in HGG tumors (Fig. 3f; Supplementary Data 4a, b). Remarkably, three clusters (WP-C1, PP-C1, and PP-C2) display similar alteration patterns: Cortex <*PDGFRA* HGG < *NTRK* HGG (Fig. 3b, c), and the majority of known glioma pathways are enriched in these three clusters (Fig. 3f), suggesting that *NTRK* HGG activates similar oncogenic pathways, but with a greater magnitude of response at the global pathway level than *PDGFRA* HGG. Moreover, the majority of HGG cancer signaling pathways are only altered in phosphoproteome but not in whole proteome (Fig. 3f), underlining the indispensable role of phosphoproteomic profiling to decode oncogenic signaling. Thus, these results suggest RTK oncogenes drive massive rewiring of signaling networks at the phosphorylation and/or protein expression level in the HGG mice.

We then investigated the global changes of regulatory protein families including TFs, epigenetic genes, kinases and cancer genes in the HGG tumors. Regulatory proteins in general are present at low abundance[32], thus difficult to analyze without highly sensitive methods. Nevertheless, our deep profiling systematically characterized both whole protein and phosphorylation levels of a large number of regulatory proteins (Supplementary Fig. 5, Supplementary Data 4c, d). Strikingly, we observed a global increase of protein expression and phosphorylation of most regulatory protein families in HGG tumors (*p* value was determined by one-way ANOVA, and cutoff 0.001 was applied, Supplementary Data 4c, d). The majority of these proteins were expressed and phosphorylated even higher in *NTRK* HGG tumors when

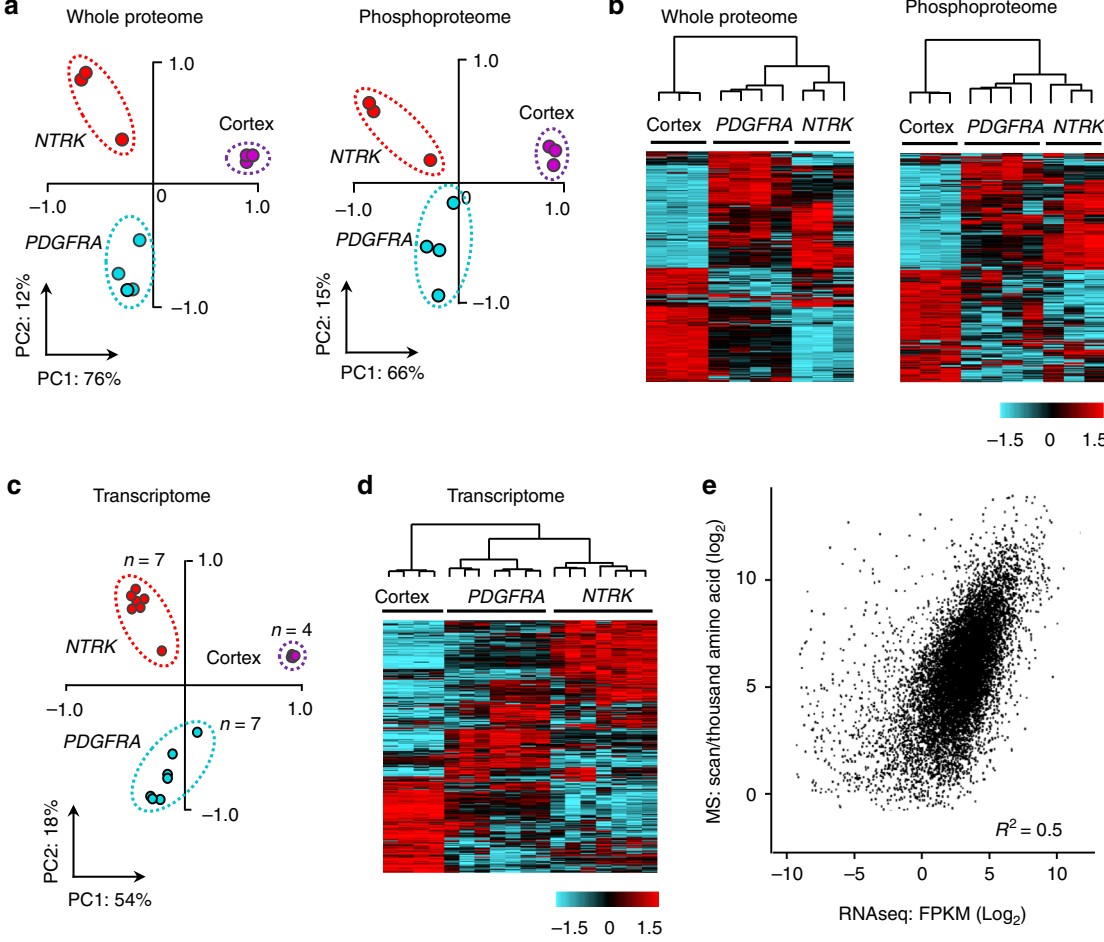

**Fig. 2** Multiomics data enable classification of the cortex, *PDGFRA* and *NTRK* samples. **a** These samples were separated by two-dimensional principle component analyses at proteome and phosphoproteome levels. The TMT intensities of all quantified proteins and phosphopeptides were log-transformed in these analyses. **b** These samples were clustered together at proteome and phosphoproteome levels by unsupervised hierarchical clustering. TMT intensities of each sample were log-transformed. The 1000 most variable proteins or phosphosites were sorted by the median absolute deviation (MAD). The hierarchical clustering was performed by WARD's method. Color key shows the $Z$ value of each protein or phosphosite. **c** These samples were separated by two-dimensional principle component analysis at the transcriptome level. FPKM values of transcripts were log-transformed. **d** These samples were clustered together at transcriptome level by the same unsupervised hierarchical clustering method in panel (**b**). FPKM values of transcripts were log-transformed. Color key shows the $Z$ value of each protein or phosphosite. **e** Moderate correlation between transcriptome and proteome abundance. Scatter plot shows $Log_2$FPKM of transcripts on *x*-axis and their corresponding $Log_2$ values of protein PSMs per 1000 amino acids on *y*-axis. PC principle component, FPKM fragments per kilobase of transcript per million mapped reads, TMT tandem mass tag

compared with *PDGFRA* HGGs. Indeed, most top DE genes show the expression pattern of *NTRK* > *PDGFRA* > Cortex (Supplementary Fig. 5a–h), including well-known master regulators (e.g. CHEK1, MAP3K1, PRKD1, INSR, and RB1) of HGG oncogenic pathways. Numerous other regulatory proteins (LYN, HMGB2, HMGA2, CD74, and CTNNB1) also follow this pattern. Lyn (Supplementary Fig. 5b) is an SRC family tyrosine kinase that enhances Glut-4 translocation to the cell membrane to increase glucose uptake[33], a hallmark of cancer metabolism[34]. HMGB2 and HMGA2 (Supplementary Fig. 5c, d) are transcription and chromatin modulators that promote stemness and tumorigenicity in HGG[35]. CD74 (Supplementary Fig. 5e) is an attractive candidate target for immunotherapy as it is present in limited amounts in normal tissues but high levels on a variety of hematological tumors[36]. CTNNB1 (Supplementary Fig. 5f) regulates cell adhesion and WNT signaling[37]. Thus, our results indicate an active role for TFs, epigenetic genes, kinases and cancer genes to reprogram signaling networks and maintain tumor homeostasis. Additionally, the *NTRK* genotype drives stronger global reprogramming than the *PDGFRA* genotype.

**Multiomics integration identifies master kinases and TFs.** Since protein kinase activity can be inferred by substrate phosphorylation levels using computer programs, we used IKAP[38], a machine learning algorithm, to evaluate the activities of 187 kinases, 41 of which are reprogrammed in the mouse HGGs (Supplementary Data 4e). Hierarchical clustering analysis classified these kinase activities into multiple major clusters (Fig. 4a), resembling three major differential regulation patterns among cortex, *PDGFRA*, and *NTRK* HGGs in Fig. 3c (i.e. PP-C1, PP-C2, PP-C5). These individual kinases were further connected in a kinase-to-kinase signaling transduction network by examining the reported kinase–substrate relationships in PhosphoSitePlus database (Fig. 4b). Multiple known kinases in gliomagenesis are identified in HGGs, encompassing AKT, PKC, MAP Kinase cascade, and SRC family kinases[11,39–41]. Other kinases regulating key intracellular systems are rewired as well, including AMPK (PRKAA1, PRKAA2) and p21-activated kinases (PAK1, PAK3). AMPK is a metabolic master sensor that regulates glucose transporter GLUT4 production, fatty acid β-oxidation, and mitochondria biogenesis[42]. PAKs regulate cytoskeleton

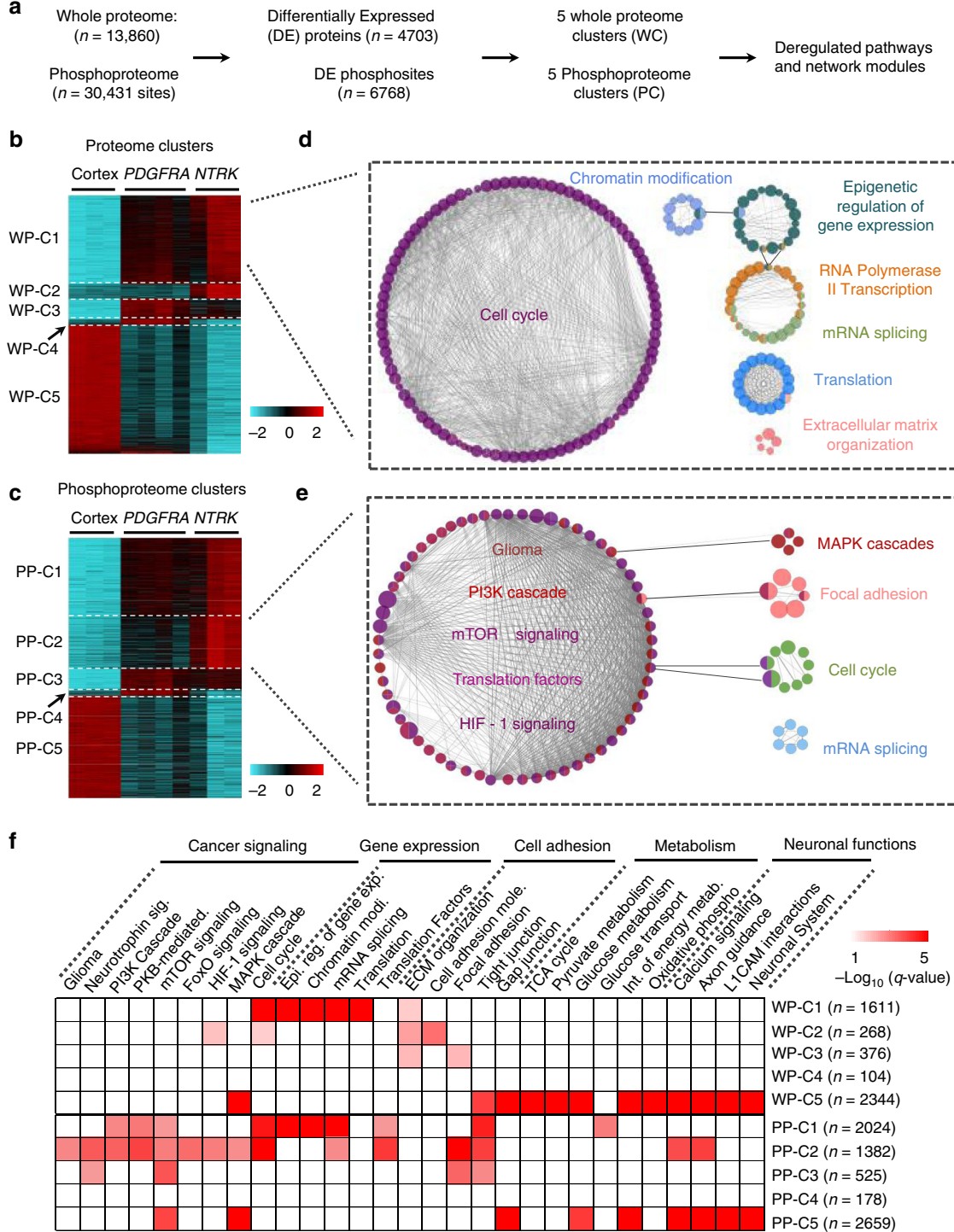

**Fig. 3** Global network analyses identify both canonical and novel network modules in HGGs. **a** Overview of the analysis strategy. Differential expression (DE) genes were selected through one-way ANOVA and clustered by WGCNA. Each coexpression cluster was utilized for pathway and network module analysis with ClueGO. **b**, **c** Multiple DE whole protein (WP) and phosphoprotein (PP) coexpression clusters with distinct patterns were detected through WGCNA analysis. Heatmaps show the z-scores of log-transformed DE WPs or PPs. WP-C whole proteome cluster, PP-C phosphoproteome cluster. The color keys present the Z scores of proteins. **d**, **e** Interconnected network modules identified in some clusters (e.g. WP-C1 and PP-C2). DE proteins/phosphorylations in WP-C1 and PP-C2 were first applied for pathway enrichment, enriched pathways were then applied for pathway to pathway network analysis. Network modules are shown in circular layout nodes and are represented by distinct colors; nodes with more than one color indicate that proteins detected in these specific pathway nodes are shared by more than one network module. Each node represents a pathway, with node size reflecting pathway enrichment significance, $p$ value was generated by Fisher's exact test, and cutoff 0.05 was applied for pathway enrichment. Functionally related pathways are connected by edges and then grouped to network modules. **f** Summary of pathways enriched in the coexpression clusters. Representative pathways detected in each cluster were organized based on biological processes. Chart shows the significance of each enriched pathway, the significance is reflected by $-\text{Log10}$ ($q$ value). Fisher's exact test was performed to obtain $p$ values, and further adjusted by Benjamini−Hochberg procedure to generate $q$ values. Cutoff of $q$ value 0.05 was applied. Number of DE proteins, phosphorylations was shown on the right side of each cluster. HGG high-grade glioma

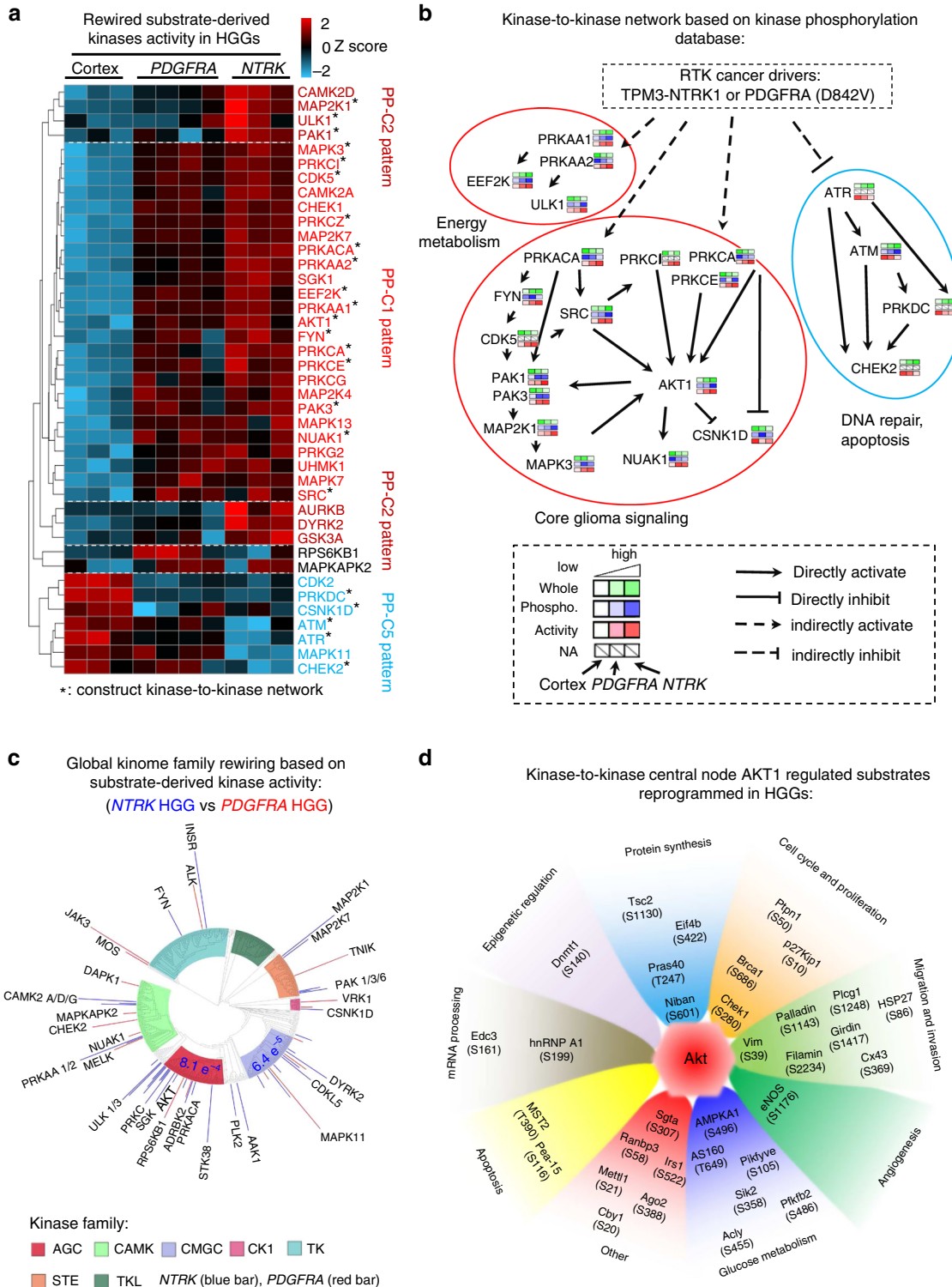

reorganization and cell motility[43]. HGGs also show higher levels of CDK5, CAMK2A, and CAMK2D, compared with normal cortex. Although these kinases are well-characterized regulators of neuronal function and synaptic plasticity, they are also expressed in glioblastoma, where they play roles in migration, invasion, mitochondrial regulation, and calcium signaling[44–46]. We further summarized the activities of these kinases at the level of kinase superfamilies. While AGC (cyclic nucleotide-dependent family, protein kinase C family, ribosomal S6 family and related kinases), CMGC (cyclin-dependent kinases, mitogen-activated

protein kinases, glycogen synthase kinases and cdk-like kinases) and CAMK (primarily kinases modulated by calcium/calmodulin) superfamily kinases are turned on significantly in HGG tumors (p value was determined by Fisher's exact test, and cutoff 0.001 was applied, Supplementary Fig. 6), *NTRK* HGGs display even higher activity in AGC and CMGC superfamilies than *PDGFRA* HGGs, supporting stronger cell proliferation signaling and cell cycle rewiring[47] (Fig. 4c).

Considering that AKT is the central node of PI3K-AKT signaling cascade upon RTK activation, we further analyzed the

**Fig. 4** Kinase activity analysis reveals active kinases, kinase superfamilies, and AKT substrates in HGGs. **a** Hierarchical clustering analysis of reprogrammed kinase activities, which were derived from substrate phosphorylation in the HGG tumors via the IKAP algorithm. The hierarchical clustering was performed by WARD's method. Activity patterns of protein clusters were compared with the phosphoproteome expression patterns in Fig. 3c. Patterns that match the clusters in Fig. 3c were labeled on the right side of the panel. For instance, cluster on the most upper side shows an activity pattern of *NTRK* > *PDGDFRA* > Cortex, similar to the phosphoproteome pattern 2 (PP-C2 pattern) in Fig. 3c. Color key represents the *Z* score of substrate-derived activities. **b** Construction of a putative core kinase-to-kinase signaling network with consistent co-activation patterns across different samples. To construct a putative kinase centered signaling network in HGG, we incorporated the relationships of kinase-substrate from PhosphoSitePlus, and manually accepted kinase–kinase networks with consistent coactivation patterns across different samples. Solid black arrows represent direct activation through phosphorylation and dotted black arrows represent indirect connection between upper kinase and downstream kinase. **c** Summarization of individual kinase activity into kinase superfamily shows stronger rewiring of AGC and CMGC superfamilies in the *NTRK* HGG than the *PDGFRA* HGG. Kinome tree map shows pairwise kinase activity comparisons, in which the kinase superfamilies with statistically significant difference are shown with *p* values determined by chi-square. Kinase activity levels are represented by the length of bars outside of the kinome tree circle. **d** Putative substrates activated by AKT, the central hub of PI3K-AKT cascades in HGGs. AKT-regulated substrates and their phosphosites are organized by annotated functions. Biological function groups are presented by distinct colors. The distance between substrates and the AKT center is correlated with substrate phosphorylation level. e.g. AMPKA1 is one of the AKT substrates with the highest phosphorylation level. HGG high-grade glioma

output of kinase activation on AKT substrates (Fig. 4d). 34 AKT substrates show a phosphorylation pattern in agreement with AKT activity (Supplementary Fig. 7, Supplementary Data 4f). The top activated AKT substrates are cell cycle and proliferation regulators (CHEK1 S280 and BRCA1 S686), central glucose metabolism regulators (e.g. AMPKA1 S496 and AS160 T649), and migration and angiogenesis regulators (e.g. eNOS S1176, VIM S39, and FLNC S2234, Fig. 4d). Similar results were obtained for the coregulation of other kinase−substrate connections (AMPKA1, CDK5, MAPK3, ATR, ATM, PAK1, and FYN, Supplementary Fig. 8). Collectively, our comprehensive kinase activity analysis enables the identification of master kinases and the downstream outcomes of kinase activation in two HGG tumor models in which PI3K pathway activation is driven by different receptor tyrosine kinase mutations that are found in human HGG.

We also explored the activity of TFs through integrative analysis of transcriptome, proteome and phosphoproteome in the mouse HGGs via a systems biology approach in multiple steps (Fig. 5a). A total of 47 TF activities were derived from target gene expression in the transcriptome, of which 38 TFs were identified by MS. Additional whole proteome and/or phosphoproteome data supported the activation of 23 out of the 38 TFs (Fig. 5b). For instance, five TFs show active status based upon the increase of phosphorylation at the reported activation sites (c-MYC S62, JUND S100, JUN S73, BRCA1 S686, and EP300 S2312). Among the most activated TFs, were the TFII family (GTF2B, GTF2F1, TAF1, TAF7, and TBP) of general TFs, which assemble the RNA polymerase II pre-initiation complex and control general transcription rate; and the transcription suppressor REST, a chromatin modifier in brain[48]. Consistently, REST target gene expression was low in the tumors and high in normal cortex (Fig. 5b), implicating a possible role of REST in HGG tumor transformation through suppression of target gene expression. Thus, this integrative analysis reveals the activation of both TF activators and suppressors, which lead to distinct reprogramming of tumor cell transcriptome and proteome.

Finally, we constructed a kinase-TF centered network in HGG by incorporating known kinase to kinase and TF to target genes connections in the PhosphoSitePlus database and the ENCODE database, with consistent coactivation patterns in our datasets, resulting in a simplified kinase-TF centered network consisting of five kinases (AKT1, MAPK3, CDK5, PRKCA, and AMPK) and six TFs (c-MYC, JUND, JUN, EP300, BRCA1, and CEBPB) (Fig. 5c). The c-MYC family (MYC, MAX) and AP-1 family (JUN, JUND) regulate a variety of central biological processes in tumorigenesis. Indeed, more than 100 c-MYC targets were

transcriptionally active, strongly supporting the central role of c-MYC in HGG tumors (Fig. 5c).

Close examination of master TFs uncovered major transcriptionally activated downstream biological processes in HGGs (Fig. 5c). The most activated biological processes are related to proliferation including ribosome biogenesis, translation, and RNA processing (targets of c-MYC, BRCA1 and CEBPB), energy metabolism including mitochondrial and metabolism (targets of c-MYC, JUN and EP300), and cell migration including extracellular matrix and focal adhesion (targets of JUN and EP300). In the kinase-TF network, c-MYC, JUN and EP300 are activated via phosphorylation by master energy and proliferation sensors kinases (AMPK and MAPK). Notably, multiple metabolic enzymes activated by these three TFs were reported to be critical during proliferation and energy stress, such as adenine phosphoribosyltransferase (APRT), which regulates a nucleotide salvage pathway to synthesize purines de novo[49], and ornithine decarboxylase (ODC1) for polyamine biosynthesis in response to growth stimulation. In summary, our systems biology approaches utilize multilayer information to prioritize central HGG TFs, kinases, and their interplay in HGG tumors.

Cross-species omics integration may be an effective approach to identify oncogenic events in human cancer[50]. Therefore, we also mapped the most significantly altered omics datasets in mouse HGGs back to human transcriptome data to search for consistent alterations driven by *NTRK* and *PDGFRA* mutations across species, resulting in a list of 20 convergent alterations in mouse and human (Supplementary Fig. 9). The majority of these genes were reported to be functional in cancer-related processes, including the regulation of cancer cell stemness, angiogenesis, tumor microenvironment, and invasion, together highlighting the potential of inter-species analysis to prioritize cancer-relevant candidates from massive multiomics datasets.

**NTRK is a stronger oncogenic driver than PDGFRA in mouse HGG.** The bioinformatics analyses suggest stronger global cancer network rewiring in *NTRK* HGG than *PDGFRA* HGG, indicating higher oncogenic potency of *NTRK* than *PDGFRA* mutations. To evaluate the oncogenic potency of the RTK cancer driver genes, we modified the PAC algorithm that was initially designed for gene expression analysis[51], to compute the summed PI3K-AKT signaling activity. The protein activity was derived from phosphosite with known functions that either promote or inhibit tumorigenesis (see Methods). In both HGGs, the PI3K-AKT pathway was clearly active and invoked similar downstream pathways, such as protein synthesis (S6, 4EBP1 and EIF4B), cell cycle progression (RB1, MYC and RBl2), cell proliferation and

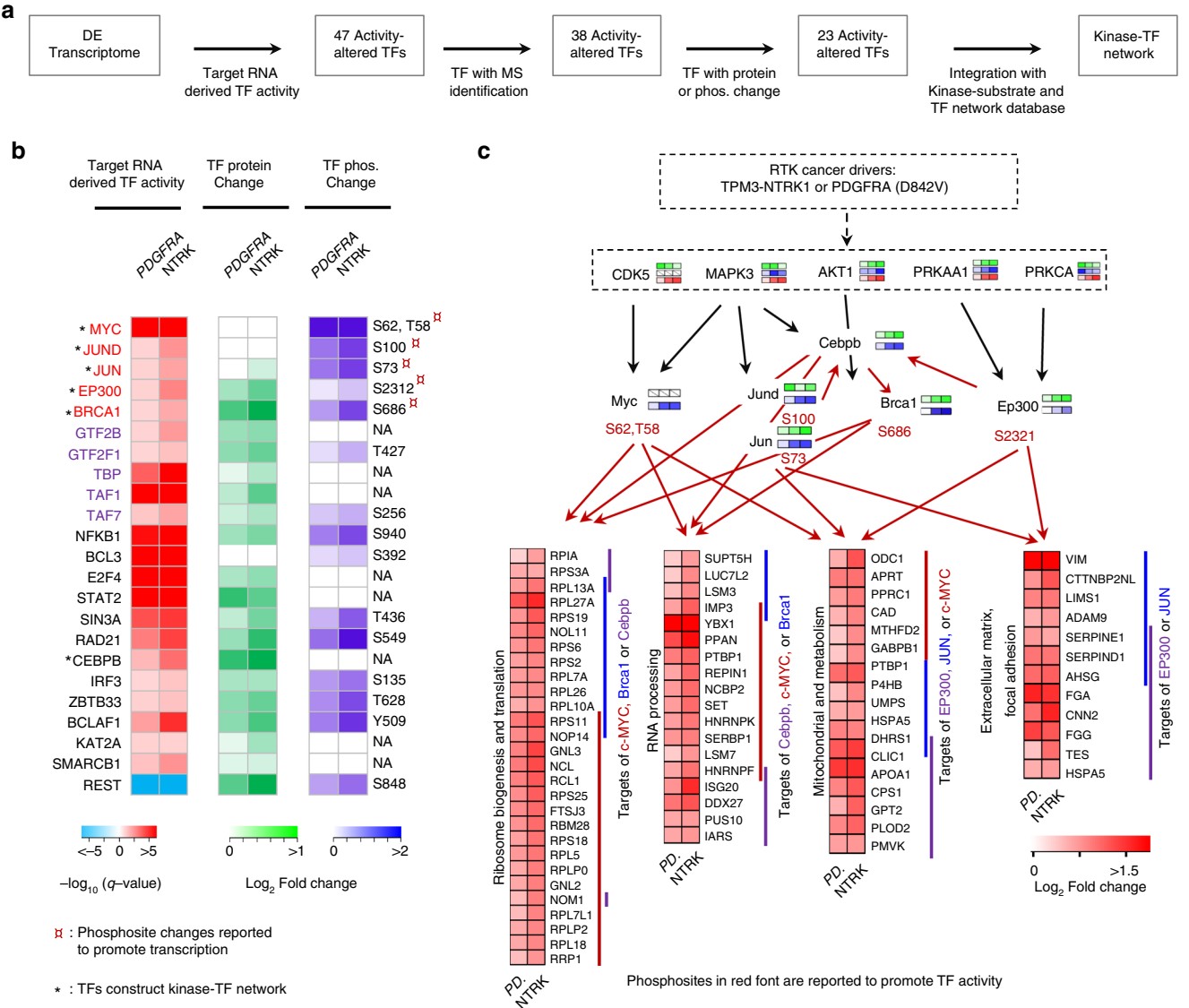

**Fig. 5** Multiomics integration identifies active TFs and constructs a core kinase-TF network in HGGs. **a** Overview of the integrative analysis strategy for TF activity inference and construction of kinase-TF network by incorporating transcriptome, whole proteome, phosphoproteome, kinase-substrate database and TF-target database. **b** Active TFs in HGGs. TF activities are first indicated by the *q* values derived from DE mRNA targets. Heatmap of MS-based quantification of the protein and phosphorylation level changes of corresponding TFs relative to cortex samples are shown in the middle and right side. Phosphosite changes that are reported to promote TF transcription are highlighted with asterisk. Font of TFII family is shown in purple. Top candidate TFs that construct Kinase-TF network through Omics data and Kinase-substrate, TF-target database is marked with asterisk. **c** Integrative analysis reveals a putative core signaling network, encompassing selected kinases, TFs, and gene targets with coactivation patterns. Dotted black arrows represent indirect connection between upper kinase and downstream kinase; solid black arrows represent direct activation through phosphorylation; and red arrows represent transcriptional activation of targeted genes. Targeted genes are organized by their enriched functional annotations. PD PDGFRA. TF transcription factor, HGG high-grade glioma

angiogenesis (BRCA1, eNOS, ERK) (Fig. 6a). When comparing 27 regulatory phosphosites of these proteins that were statistically different between *NTRK* and *PDGFRA* HGGs, the majority (*n* = 23) showed higher alteration in *NTRK* HGG than *PDGFRA* HGG (Fig. 6b; Supplementary Data 4g). Consistently, the *NTRK* HGG exhibited 1.45-fold greater PI3K-AKT signaling activity (*P* = 0.003, Supplementary Data 4h), suggesting that the *TPM3-NTRK1* fusion gene harbors stronger oncogenic potency than the *PDGFRA* D842V.

To experimentally validate our predicted oncogenic potency of *TPM3-NTRK1* and *PDGFRA* D842V, we analyzed cellular proliferative indexes and Kaplan−Meier survival curves of both HGG mice. The proliferative index was defined by the proportion

of tumor cells that expressed the proliferation marker Ki67 (Fig. 6c). We quantified the proliferative indexes in four *NTRK1*-driven HGGs and four *PDGFRA*-driven HGGs with enhanced level of PI3K-AKT signaling, the proliferative index of *NTRK* HGG (0.32 ± 0.04) was 1.45-fold higher than that of *PDGFRA* HGG (0.22 ± 0.07), with a *p* value of 0.048 determined by Student's *t* test. Consistently, *NTRK* HGG mice developed with much shorter latency than *PDGFRA* mice (median survival time of 16 days and 30 days, respectively, Fig. 6d).

*TPM3-NTRK1* and *PDGFRA* D842V both activated PI3K-AKT signaling, but with different potency. Species-specific analysis of PDGFRA protein identified a difference in RTK upregulation between the two HGG models. MS measurements showed that

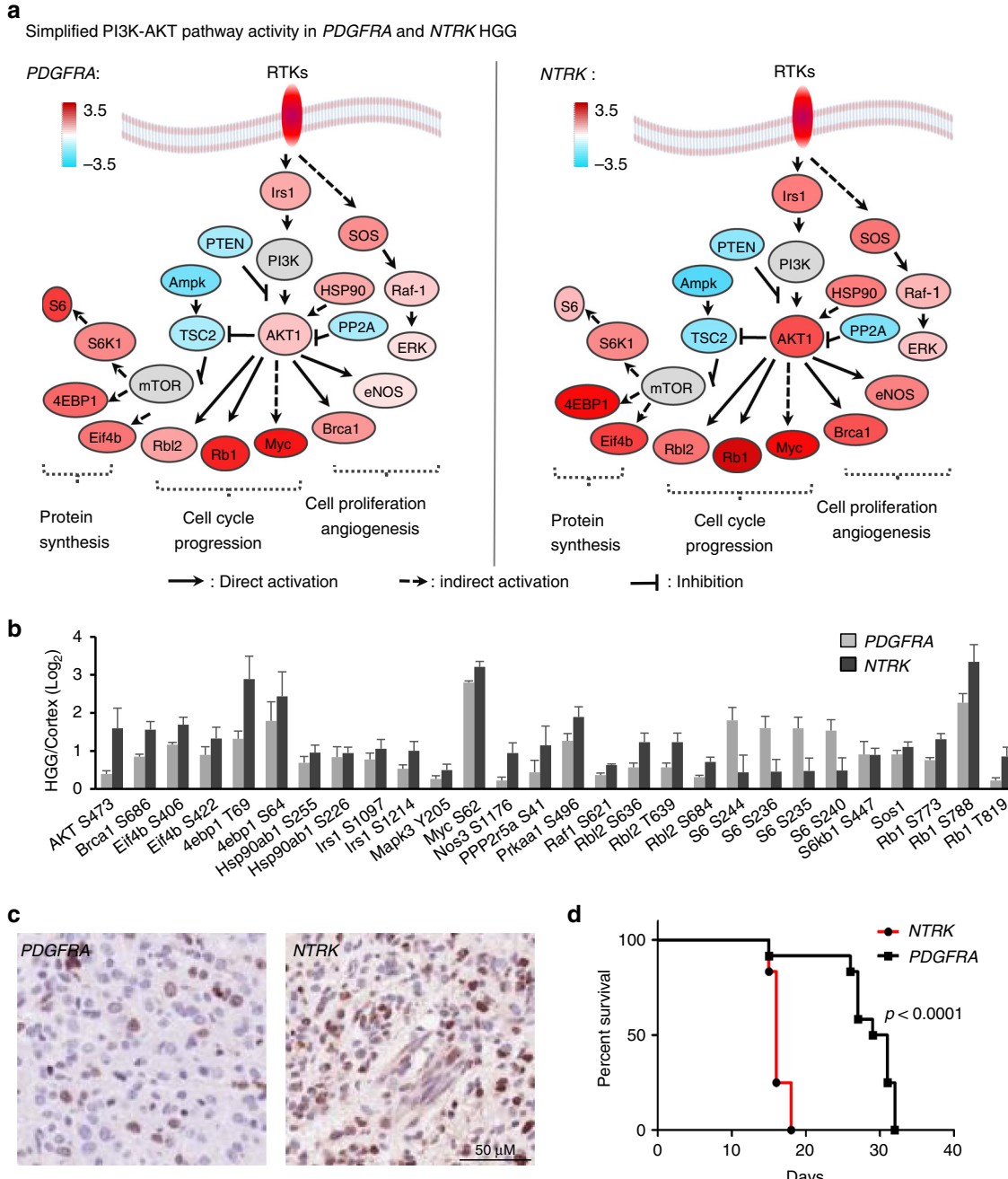

**Fig. 6** *NTRK* fusion is a stronger oncogenic driver than *PDGFRA* mutation in mouse HGG. **a** PI3K-AKT pathway is active in both *PDGFRA*-driven and *NTRK*-driven HGGs. Simplified PI3K-AKT pathway diagram shows activity change according to the phosphorylation of functional sites. Color key shows the Log$_2$ fold change of phosphosites in *PDGFRA*- or *NTRK*-driven tumors compared to normal cortex. **b** *NTRK*-driven HGG displays stronger PI3K-AKT signaling activity than *PDGFRA*-driven HGG. MS-based measurements of phosphosites with activation or inhibition functions. Error bar indicates s.e.m. **c** *NTRK*-driven HGG shows higher cell proliferative index compared to *PDGRA*-driven HGG. Representative Ki-67 IHC-stained sections on *PDGFRA*- and *NTRK*-driven HGGs to examine the proliferative indexes of HGGs. A 50 μm scale bar is shown. **d** *NTRK*-driven HGG exhibits short tumor onset latency than *PDGFRA*-driven HGG. HGG K−M curve confirms more rapid mice tumor onset of *NTRK*-driven HGGs (*n* = 12) compared to *PDGFRA*-driven HGGs (*n* = 12). The *p* value was determined by Student's *t* test. HGG high-grade glioma, MS mass spectrometry

HGGs driven by the human *PDGFRA D842V* gene expressed lower levels of the mouse wild-type PDGFRA protein than the *NTRK* HGGs (Fig. 7a). We quantified endogenous mouse PDGFRA by mouse amino acid sequence-specific peptides, and validated this finding by Western blotting (Fig. 7b). Many other RTKs (EphA2, EGFR, FLT4, PTK7 and ROR2) also showed higher protein expression in *NTRK* HGG than *PDGFRA* HGG (Fig. 7c). Transcriptomic measurement consistently indicated the

upregulation of these RTKs. Western blotting further confirmed EphA2 overexpression and activation reflected by concomitant phosphorylation (Fig. 7d). To identify potential mechanisms driving increased RTK expression, we searched ENCODE and MsigDB databases to identify TFs that regulate the RTK transcription and validated TF activities by their protein levels or phosphorylation states (Fig. 7e). We used Western blotting to confirm increased expression and phosphorylation of MYC as a

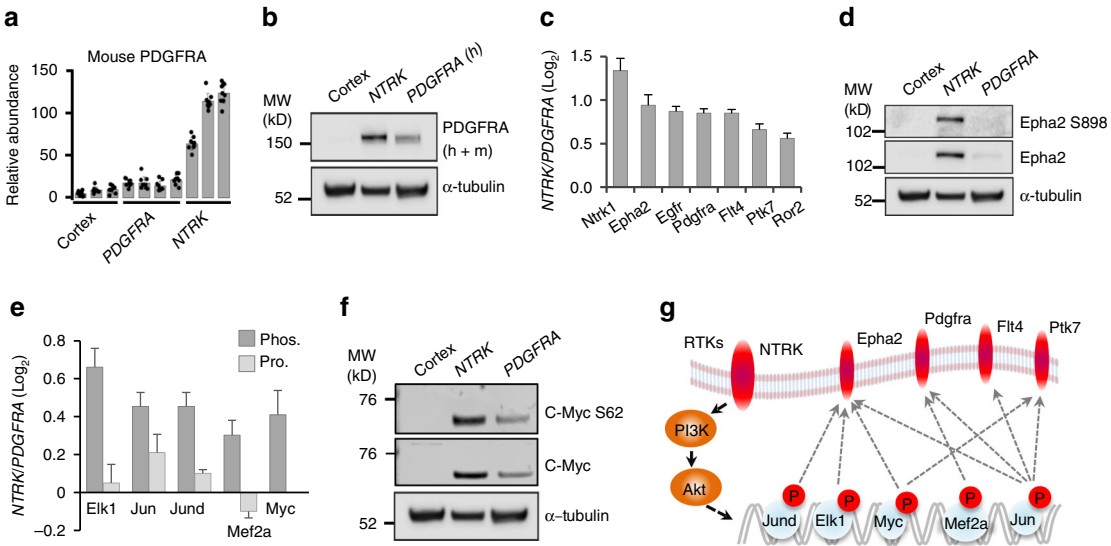

**Fig. 7** *NTRK* enhances the expression and activation of other RTKs. **a** *NTRK* fusion gene induces overexpression of endogenous mouse PDGFRA in *NTRK*-driven HGG. Mouse and human PDGFRA peptide sequences were first compared. Mouse amino acid sequence-specific PDGFRA peptides in the MS analysis were pulled out for quantification. Error bar indicates s.d. **b** Immunoblot assay on whole PDGFRA expression (h: human; m: mouse). **c** *NTRK* fusion gene upregulates multiple other RTKs in *NTRK*-driven HGG as measured by mass spectrometry. *p* value was determined by Student's *t* test, and cutoff 0.05 was applied. Error bar indicates s.e.m. **d** Immunoblot assay on EPHA2 protein expression and phosphorylation (S898). **e** TFs that promote RTK expression are more active in *NTRK*- than PDGFRA-driven HGGs. Bar plot shows MS measurements of TF activation phosphosites and protein expression levels. MS measurement of Myc protein level was missing in the data. Phos. phosphorylation level, Pro. protein level. Error bar indicates s.e.m. **f** Immunoblot assay validates the activation of c-Myc. Western blotting was performed for c-Myc protein and c-Myc S62 phosphorylation. **g** A positive feedback loop model in the *NTRK*-driven HGG. Red ovals and circles indicate enhanced expression or activation of RTKs or transcription factors induced by *NTRK* fusion in mouse HGG. Solid arrow indicates direct regulation, dot arrow indicates indirect regulation. MW molecular weight, RTK receptor tyrosine kinase, HGG high-grade glioma, MS mass spectroscopy

representative TF (Fig. 7f). Together, these bioinformatics and experimental findings demonstrate that the *NTRK* fusion gene induced an enhanced overexpression and activation of other RTKs, suggesting a forward feedback loop within PI3K-AKT signaling, resulting in a more aggressive tumor than *PDGFRA*-driven HGG (Fig. 7g).

**CRISPR-Cas9 screening validates molecular targets in HGG.**
To examine if master regulators prioritized by the multiomics approaches are required for tumor survival, we first established the in vitro culture of HGG primary cells collected from NTRK-driven HGG mouse tumor tissues, and then designed a pooled mini-gRNA library for targeting these master regulators in a CRISPR-Cas9 analysis (Fig. 8a, Supplementary Fig. 10). We used a TransEDIT-dual CRISPR-Cas9 system[52], in which recombinant lentiviruses expressed dual gRNAs designed by a machine-learning approach to promote the functional ablation of genes (Supplementary Fig. 10b). Six nontargeting gRNAs were included as negative controls (Supplementary Data 4i). Systematic experimental optimizations were performed (Supplementary Fig. 10a), for example, stable expression of Cas9 in the HGG cells was confirmed by immunoblotting (Supplementary Fig. 10d); gRNA integration was validated by fluorescence detection of ZsG (Supplementary Fig. 10e); relatively even distribution of each gRNA in the pooled library was confirmed by deep-sequencing before screening (Supplementary Fig. 10g), and screening was performed in triplicate for reproducibility.

We targeted two types of master regulators (kinases and TFs) during the CRISPR-Cas9 screening, including nine genes derived from transcriptome and proteome data (Fig. 8b), and we also targeted six genes identified by cross-species comparisons with mouse and human tumors, each with six different gRNAs (i.e. three different dual-gRNA constructs). Dropout analysis was

conducted to identify the essential regulators responsible for tumor survival. If two out of the three dual-gRNA counts were significantly decreased after selection for 15 days compared to those in starting populations, the targeted gene was regarded to be important for tumor viability. Under this cutoff, none of the negative control gRNAs were enriched. On the other hand, 56% (5 out of 9) of the prioritized master regulators were shown to be critical for the HGG tumor growth. Strikingly, all three kinases (i.e. PRKAA1, PRKAA2, and EEF2K) regulating cell metabolism were found to contribute to HGG cell viability, providing a novel tumor vulnerability. Moreover, two TFs (Jun and Myc) were demonstrated to be positive hits in the screening. Given that RTK-PI3K-AKT induces a broad spectrum of downstream changes, pinpointing out Jun and Myc leads to valuable insights on how RTK fusions induce HGG tumorigenesis. Thus, this CRISPR-Cas9 screening unveils a novel tumor vulnerability of energy metabolism, and the involvement of Jun and Myc in NTRK fusion-induced tumorigenesis, together confirming the validity of the multiomics integrative approach to discover master regulators.

**Discussion**
Both transcriptomic and proteomic analyses play indispensable roles for understanding the underlying central regulatory mechanisms in cancer biology. As mRNA level is often only moderately correlated with protein level[29], there is a need to profile both transcriptome and proteome to obtain a full picture of gene expression in cancer biology. Here we demonstrate the power of deep proteome coverage and integration of multiomics datasets to probe molecular mechanisms underlying tumorigenicity. A pooled mini-gRNA library CRISPR-Cas9 functional screening further validates the strength of the multiomics integrative approaches and identifies multiple tumor vulnerabilities

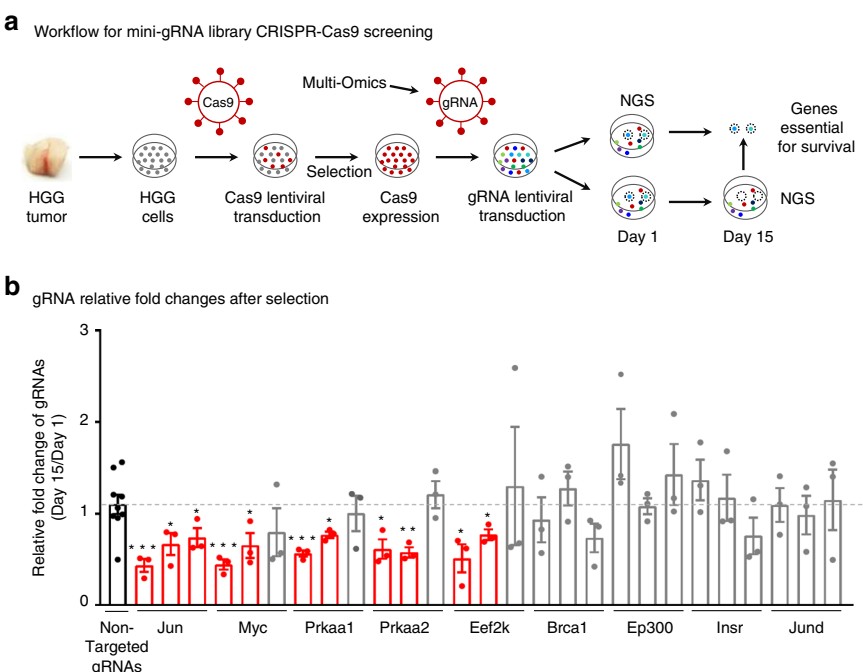

**a** Workflow for mini-gRNA library CRISPR-Cas9 screening

**b** gRNA relative fold changes after selection

**Fig. 8** CRISPR-Cas9 validation screening identifies novel candidate tumor vulnerabilities in HGG. **a** Workflow for customized mini-gRNA library CRISPR-Cas9 functional screening using transEDIT-dual system. Primary HGG tumor cells were generated from HGG tumor tissue, Cas9 lentivirus was transduced to primary cells and selected by antibiotics blasticidin to produce Cas9 stable expression cells. Pooled mini-gRNA library was then transduced and evenly split to two aliquot after 24 h of transduction. The first aliquot was collected and stored as pool start and the other stock was selected for 15 days and then collected as pool final for Miseq sequencing. Genes essential for cell viability were calculated by comparing gRNA count difference between pool final and pool start. NGS next-generation sequencing. **b** CRISPR-Cas9 functional screening targeting top master regulators identifies genes for HGG viability, confirming the validity of the multiomics integrative analyses. Bar pot shows gRNA count changes of top kinases and transcriptional factors identified through the multiomics integrative analyses comparing pool final counts (Day 15) to pool start counts (Day 1). Dash line indicates the mean level of all nontargeted gRNA controls. The gRNA counts of master kinases and TFs were compared with nontargeted gRNA controls, and red bars indicate gRNA changes that show statistical significance, gray bars show no statistical significance, black bar shows the values of nontargeted gRNAs. Error bars indicate the s.e.m. of three experiments. $p$ values were determined by Student's $t$ test. One asterisk: $p < 0.05$, two asterisks: $p < 0.01$, three asterisks: $p < 0.001$. HGG high-grade glioma, TF transcription factor

for designing targeted cancer therapy. Recent developments of optimized long gradient LC-MS/MS system[27], refined phospho-peptide enrichment[26], and advanced bioinformatics tools[7,25] greatly improve the depth of proteomic profiling for cancer studies, facilitating detection of almost all of the expressed proteins. Such high coverage allows the systematic analysis of proteins of low abundance, as exemplified by TFs and kinases. In parallel, a comprehensive phosphoproteome analysis offers complementary information about pathway/network activities, because many components in pathways are not changed at the protein level, but altered in phosphorylation states during signaling transduction. Although some known phosphosites are missed due to intrinsic limitations of the unbiased proteomics approach, we have detected nearly all NTRK and PDGFRA regulated pathways in the KEGG database in this deep phosphoproteome dataset.

Isobaric labeling (e.g. TMT and iTRAQ) is a powerful quantitative strategy for multiplexed deep proteomic profiling with high throughput and reproducibility[1]. Although quantitative ratio compression often occurs with this method[53,54], it also reduces experimental variations, and therefore has almost no impact on differential expression analysis after scale normalization[1] (Supplementary Fig. 11). Moreover, our strategies of extensive peptide separation[55] with biological replicates facilitate statistical inference and largely reduce the effect of ratio compression.

Recurrent mutations in the RTK/RAS/PI3K signaling axis occur frequently in virtually all adult glioblastomas, more than half of pediatric glioblastomas, and other diverse tumor types[11,12,14]. While this implies that the PI3K pathway is an important therapeutic target, the response to small molecule inhibitors of the pathway is highly variable and often difficult to predict, likely due to varied consequences of specific mutations within the pathway, combinatorial effects with co-occurring mutations, complex feedback regulation within the pathway and cross-talk with other signaling pathways. In the present study, we investigated the sensitivity of integrated analysis of multiomics datasets to identify shared downstream pathways and differences in signaling in HGGs driven by two different glioma-associated RTK mutations in the same p53-null primary astrocyte population.

We present a generic bioinformatics pipeline for prioritizing core signaling networks and master regulators in cancer proteomics studies. Massive reprogramming of molecular components occurs during the evolution from mortal to immortal status in cancer cells[3]. As improvement of profiling technologies allows the identification of thousands of these changes, strategies for prioritizing drivers and core regulators from the enormous amount of passenger changes become urgently needed. Here, we first performed weighted coexpression clustering analysis to extract ten proteome and phosphoproteome clusters from 4703 differentially expressed proteins and 6768 differentially phosphorylated phosphosites, which dramatically reduced the data complexity. This readily identified major pathways with well-established roles in glioma growth as well as clear connections with PI3K and mTOR signaling downstream of RTK activation[12,17]. Subsequently, coregulated genes in each of the clusters were summarized to pathways and networks using the

network analysis method, which further narrowed down these massive changes to 67 network modules. We also developed systematic protein activity inference strategies for kinases and TFs by integrating multiomics data and a variety of databases to further prioritize 41 kinases, 23 TFs, and a core network consisting of 9 master regulators from these gene clusters and network modules.

Importantly, this integrated approach extended beyond simple identification of pathways to illustrate differences between the two oncogenes. Even within a similar cellular context in normal tissues, unique aspects of signaling downstream of different RTKs are required to transduce distinct functional consequences driven by different ligands. Oncogenic *NTRK* fusion genes are found at low frequencies across many different types of cancer[56]. In brain, NTRK fusions may be found in adult or childhood HGGs carrying multiple other mutations, and they are also found in infant HGGs and childhood low-grade gliomas in which very few nonsynonymous mutations can be detected[11,13,21,57,58]. Differences in biological settings and concurrent mutations further complicate the ability to directly compare downstream effects of different RTK mutations in human tumors. Using model systems to compare two different glioma-derived RTK mutations in a more uniform setting revealed a higher amplitude of pathway activation and a feed-forward upregulation of other RTKs in NTRK-driven HGGs compared with PDGFRA-driven HGGs. This is consistent with the higher proliferative index and shorter tumor latency of the mouse HGGs, and the ability of NTRK fusions to act as potent oncogenic drivers in primary human tumors with minimal co-occurring mutations.

A CRISPR-Cas9 functional screen to validate essential downstream regulators showed a critical role for regulators of energy metabolism. *NTRK* HGG cells showed dependencies on two different isoforms of the catalytic alpha subunit of AMPK, PRKAA1 and PRKAA2. AMPK, a key regulator of energy balance and autophagy, is a heterotrimeric complex with diverse composition yielding 12 different potential complexes based on multiple isoforms for each of the three subunits. The selective difference between different AMPK complexes is not well understood[59]. Dependencies on two different alpha subunits show that there is insufficient AMPK redundancy in NTRK HGG cells to tolerate loss of either of two AMPK complexes. Consistent with this finding, during the revision of this manuscript, AMPK inhibition was reported to reduce viability of HGG tumors, supporting the potential of AMPK pharmacological inhibitors as a novel therapeutic strategy for HGG[60]. Interestingly, *NTRK* HGG cells were also dependent on EEF2K, a negative regulator of elongation factor 2. EEF2K activity slows protein synthesis, increasing the cellular pool of amino acids and energy, and possibly favoring selective translation of specific proteins[61]. AMPK and EE2FK play complex roles in cancer. They are important therapeutic targets, as they help cancer cells overcome cellular stresses such as nutrient deprivation and energy depletion. However, activation of these pathways inhibits tumor growth in some settings[61,62]. Therefore, an improved understanding of context-specific effects of AMPK and EE2FK is essential to identify relevant settings for use of selective inhibitors in cancer therapeutics. Dependency screens in diverse models should provide increased clarity.

Considering the enormous complexity of genomic background in patient samples, mapping the most significantly altered omics changes in mouse models back to human data may be an effective approach to pinpoint biological mechanisms driven by specific oncogenes in patients. CRISPR-Cas9 screening was also designed to target on six genes (i.e. Cd93, Cd74, Epha2, Spry1, Arhgap18, and Dab2) prioritized through the cross-species omics data integration. Cd93 was critical for the growth of *NTRK*-driven

HGG primary cells (Supplementary Fig. 10h). Other candidates may have failed to be enriched from the screening because of low cutting efficiency of all gRNAs against the same gene. However, there were several limitations for our cross-species integration: (i) highly limited patient sample availability (i.e. only eight patients with *PDGFRA* mutations and three patients with *NTRK1* fusions) restricted the statistical power; (ii) large variations in the patients (e.g. tumor cells of origin, tumor growth environment, patient age and tumor grade) also confounded the data integration. Nevertheless, this approach would be useful for omics studies with large numbers of human samples.

With rapid improvement in omics technologies and accumulation of big datasets, this integrated bioinformatics pipeline provides a general platform for prioritizing master genes and core signaling networks in cancer omics study. Combining with CRISPR-Cas9 functional screening, the pipeline will provide enhanced mechanistic understanding of the oncogenic process and illuminate potential therapeutic vulnerabilities.

## Methods

**Mutated RTK-driven HGG mouse models and tissue collection**. Mouse experiments were approved by Institutional Animal Care and Use Committee and are in compliance with national and institutional guidelines. At passage one, a pooled population of p53-deficient primary mouse astrocytes was transduced with retrovirus expressing either human *TPM3-NTRK1* fusion or *PDGFRA D842V* mutation along with IRES-GFP, and then $2 \times 10^6$ cells per mouse were intracranially implanted into athymic nude mice for tumorigenesis before six passages[13,22]. Mice were anesthetized and perfused with PBS on the manifestation of brain tumor symptoms. Dissection of the focal regions of GFP-labeled HGG tumors was aided by visualizing with a fluorescence dissecting microscope to maximize tumor purity and minimize contaminating normal mouse cells. Dissected tumor tissue was snap frozen for proteome and transcriptome analyses. Reproducible intragroup proteome, phosphoproteome, and transcriptome signatures showed that noise introduced by low-level contamination of normal cells did not mask robust differential signatures.

*Antibodies and other reagents*: Antibodies against the following proteins were used for Western blotting at a concentration of 1:1000:p-c-Myc (Abcam, 32029); Tubulin and PDGFRA (Santa Cruz, 23948, and 338); EphA2, pEphA2, and c-Myc (Cell Signaling, 6997, 6347, and 9402). Other reagents included phosphatase inhibitors (Roche); Lys-C (Wako); Trypsin (Promega); TiO$_2$ beads (GL Sciences); and C$_{18}$ 1.9 µm resin (Dr. Maisch GmbH). IHC was performed with anti-Ki67 antibody (Novocastra, NCL-L-Ki67-MM1), and quantified using a three-step image processing method[63]; briefly, slides were scanned and processed, tissue was classified, IHC marker was detected, and an IHC staining index was calculated.

**RNAseq analysis**. RNAs were extracted by Trizol (Invitrogen) from approximately 20 mg of tumor sample aliquots that were also used for proteomic profiling. The mRNA samples were purified by poly(dT) beads, converted to cDNA followed by fragmentation and ligation with paired end adaptors. RNAseq reads were aligned to multiple databases encompass human genome (GRCh37), human transcriptome (RefSeq and AceView), and all other possible combinations of RefSeq exons. The reads mapped to the transcriptome were converted to genomic mapping and merged in the final BAM files[13].

To compare the expression levels of the human *PDGFRA D842V* and *TPM3-NTRK1* transgenes expressed in the mouse tumors with expression of PDGFRA or NTRK1 in human tumors, mouse RNAseq data were remapped to a single reference combined with human hg19 and mouse mm9 genome using STAR_2.4.1d_modified. HTseq 2.7.2 was used to quantify reads unique to the mouse and human genome. The human and mouse gene counts were then normalized to FPKM. To normalize the FPKM expression between mouse and human tumors, we identified orthologous genes and performed a quantile normalization to ensure gene quantification for both species follow the same distribution.

**Deep proteomic profiling by reverse phase LC/LC-MS/MS**. Whole proteome and phosphoproteome analyses were processed using an established protocol[5]. Tissue sample aliquots (10 mg each) were homogenized at 4 °C in 0.3 mL of lysis buffer (50 mM HEPES, pH 8.5, 8 M urea, 0.5% sodium deoxycholate, 1× PhosStop Phosphatase Inhibitors). Cell lysate including insoluble debris was digested with Lys-C (a substrate-to-enzyme ratio of 100:1, w/w) followed by trypsin (a substrate-to-enzyme ratio of 100:1, w/w) overnight at room temperature. Peptides of each sample were labeled with TMT10-plex reagents and then equally pooled. The pooled peptides were pre-fractionated with a 2 h gradient basic pH reverse phase LC, resulting in a total of 30 whole proteome peptide fractions after concatenation, and each fraction was further analyzed by long gradient (up to 9 h), acidic pH

reverse phase LC-MS/MS (Q Exactive, Thermo Scientific). MS raw data were processed using the reported JUMP software suites to improve sensitivity and specificity, which combines the advantages of pattern matching with de novo sequencing during database search[7,25]. Briefly, raw MS files were searched against the mouse database downloaded from Uniprot (52,490 entries) with Met oxidation as dynamic modification. Search parameters were precursor and product ion mass tolerance (6 and 10 ppm, respectively), fully tryptic restriction, two maximal missed cleavages, static TMT modification (+229.162932 Da on N-termini and Lys residues), dynamic Met oxidation (+15.99492 Da), three maximal dynamic modification sites, and the consideration of a, b, and y ions. Peptide-spectrum matches (PSMs) were filtered by seven minimal peptide length, mass accuracy (~3ppm) and matching scores (JUMP Jscores and dJn values) to achieve 1% protein FDR. During the quantification, the percentage of precursor peak intensity was set to at least 70%; minimum and median TMT channel intensities were set to 2000 and 10,000, respectively to guarantee only high-quality PSMs were used for quantification.

*Phosphopeptide enrichment and LC/LC-MS/MS*: The vast majority of TMT-labeled peptides were used for phosphoproteome analysis with TiO$_2$-based phosphopeptide enrichment[26]. After the basic pH reverse phase LC separation, fractions were first concatenated into a total of 20 phosphoproteome fractions, and phosphopeptides in each concatenated fraction were then enriched by TiO$_2$ beads (a peptides-to-beads ratio of 1:4, w/w) with 0.5 mM KH$_2$PO$_4$ competitor, and then subjected to acidic pH reverse phase LC-MS/MS. Data processing was essentially the same as that of the whole proteome analysis, except the inclusion of dynamic Ser/Thr/Tyr phosphorylation (+79.96633 Da). During the quantification, the percentage of precursor peak intensity was set to at least 50%; minimum and median TMT channel intensities were set to 1000 and 5000, respectively.

We determined phosphosite reliability by the localization score (Lscore) from the JUMP software suite based on the concept of the phosphoRS algorithm[64]. We first derived an Lscore (0−100%) for each phosphosite in every PSM, and then aligned all phosphosites to protein sequences to produce a protein Lscore for each phosphosite. When one site was identified by numerous PSMs, the highest PSM Lscore was selected. As random assignment often occurred for ambiguous phosphosites, inflating the number of protein phosphosites, we used multiple rules to address this issue: (i) For ambiguous phosphosites in a PSM (e.g. the gap of the first and second PSM Lscores < 10%), we searched the phosphosite information in the corresponding protein to define the site, which enabled the PSMs of low quality to borrow information from the PSMs of high quality. (ii) If neither the PSM Lscore nor the protein Lscore was distinguishable, we used a heuristic order to assign the phosphosite: SP-motif, S, T and Y. (iii) If none of the rules were applicable, we sorted the PSMs by JUMP Jscores to select the phosphosite. Principal component analyses and hierarchical clustering were performed using R (version 3.0.1). All quantified proteins and phosphopeptides were applied for the analyses. Missing values were filtered out during protein/phosphopeptide quantification, and thus were not considered in PCA and hierarchical clustering analyses.

The kinase activity should be a direct measure of the signal contributed by the phosphorylation status; therefore, we performed the whole proteome normalization. For the pathway analysis, we did not normalize to whole proteome because the protein quantity is also an integral part of the measured pathway activity. Regardless, we did not find significant differences between the normalized vs. un-normalized phosphorylation result. Un-normalized were in Supplementary Data 2-1; Normalized phosphorylation result is in Supplementary Data 2−2.

*Evaluation of proteomic profiling depth*: For whole proteome analysis, we compared the identified peptides to all theoretically observable peptides, denoted as proteotypic peptides. The proteotypic peptides were estimated by filtering in silico tryptic peptides with three parameters, including the detection in transcriptome[65], compatible peptide mass range and hydrophobicity[66]. These cutoffs were selected on the analysis of all identified peptides/proteins.

For phosphoproteome, we compared our data to all mouse phosphosites collected in the PhosphoSitePlus database, in which phosphosites sequenced by at least two independent MS analyses were accepted. Differential expression analyses of whole proteome and phosphoproteome

The analysis was performed by one-way ANOVA-based comparison of cortex, *NTRK* HGG, and *PDGFRA* HGG with *p* values estimated by permutation for 1000 times following the Storey's procedure[67] and then adjusted by the Benjamini−Hochberg method. An initial *p* value cutoff of 0.05 was applied. Remaining proteins were further filtered by a fold change of 1.5 in at least one comparison among the three groups; a final FDR was estimated by permutation using the resulted DE genes. DE phosphosites were identified by the same procedure except that the fold change was set to 2.

*Global pathway and network module analysis*: The DE proteome and phosphoproteome were analyzed by WGCNA[30] coexpression clustering analysis. Briefly, Pearson correlation matrix was calculated using all samples, allowing only positive correlation. Hybrid dynamic tree-cutting method with a minimum height for merging modules at 0.15 was applied to define coexpression clusters. The first principal component (i.e. eigengene) was calculated as a consensus trend for each coexpression cluster. DE proteins were assigned to each cluster based on Pearson *r* values.

Pathway and network module analyses were carried out using ClueGO[31], a software package based on cytoscape, for DE proteins and phosphosites in each coexpression cluster. In ClueGO, pathway analysis was performed using right-sided

hypergeometric test, with a BenjaminiHochberg corrected *p* value cutoff of 0.05. Then Kappa statistics was applied to link deregulated pathways to construct network, with a Kappa score cutoff of 0.5 to ensure stringency. The pathway information was pooled from the KEGG, WikiPathways, and Reactome databases.

*Kinase activity analysis by multiomics integration*: Kinase activity analysis was carried out using IKAP[38], a heuristic machine learning algorithm to infer kinase activities from the related substrate phosphorylation levels. Kinase−substrate relationship was extracted from the PhosphoSitePlus database. The level of substrate phosphorylation may be attributed to the related kinase activates and the abundance of substrate itself. To eliminate the effect of substrate abundance, the phosphoproteome data were normalized against the whole proteome level. In the IKAP analysis, we repeated the simulation process for ten times to overcome limitation of gradient descent optimization algorithm that could get stuck in a local minimum. Then we applied a cutoff of 0.2 standard deviation to filter results that failed to converge into a stable solution. Kinase activities derived from substrate size <3 were filtered out except ones supported by upstream kinases with coactivation patterns. Finally, we applied a cutoff of Benjamini−Hochberg-corrected *p* value 0.05 to determine the list of kinases with altered activity.

*TF activity analysis by multiomics integration*: TF activity was derived from target genes expression in transcriptome and whole proteome clusters and was further validated by the measurements of TF whole protein and phosphorylation. DE proteins from each coexpression cluster were first overlapped with TF targets to derive differential TF activities, according to TF−target relationship in the ENCODE database. ENCODE database contains only experimentally validated data, and was therefore used in this study[68]. Fisher exact test was used to determine the significance, followed by Benjamini−Hochberg correction. *p* value cutoff was set to 0.05. Similarly, we overlapped DE genes in transcriptome with TF targets to evaluate TF activities. We only accepted TFs that showed activity changes from both whole proteome data and transcriptome data. The change of these TF activities was further validated by the alteration in whole proteome and/or phosphoproteome.

Finally, to construct a putative kinase-TF network in HGG, we incorporated the relationships of kinase−substrate and TF−target from PhosphoSitePlus and ENCODE databases, respectively, and manually accepted kinase-TF networks with consistent coactivation patterns across different samples. For instance, AKT1 kinase showed different activity levels in three samples in an order of NTRK > PDGFRA > Cortex. The AKT1 is known to phosphorylate Brca1 at S686 residue. The phosphorylated level of S686 also followed an order of NTRK > PDGFRA > Cortex. Furthermore, Brca1-depenent transcripts were also elevated in an order of NTRK > PDGFRA > Cortex. Thus, we accepted the AKT1-Brca1 network.

**Combination of mouse and human HGG data**. Since we demonstrated the distinct oncogenic potency of the two RTK cancer drivers in mice, oncogene-responsive changes were restricted to gene expression patterns that correlated with the distinct oncogenic potency of the two RTKs. Thus, mouse genes with a transcript expression pattern (*NTRK* > *PDGFRA* > Cortex) was extracted with high stringency (at least two-fold between *NTRK* HGG and *PDGFRA* HGG, *p* value was determined by Student's *t* test, and cutoff 0.05 was applied). Moreover, the genes without the consistent change in neither whole proteome nor phosphoproteome in mouse HGGs were further filtered out. Similarly, we extracted human genes that had higher transcript levels in *NTRK* fusion cases than *PDGFRA* mutation cases. Finally, the mouse and human gene lists were overlapped for convergent oncogene-responsive changes.

*Pathway activity measurement using functional phosphosites*: Most of pathway activity inference strategies rely on gene expression at transcript level or protein level, which may not accurately indicate protein activity if highly regulated by phosphorylation. We modified a pathway activity inference strategy[51] to compute PI3K-AKT pathway activity in different samples, termed as a(P), based on the altered phosphoproteins with annotated functional phosphosites:

$$a(P) = \sum_{i=1}^{k} C_i \times F_i / \sqrt{k} \qquad (1)$$

in which *K* is the number of proteins with different activity relative to normal cortex samples, only PI3K-AKT pathway proteins with annotated functional phosphosites changes were accepted; $F_i$ is the averaged Log$_2$ fold change of DE phosphosites in protein *i*; $C_i$ is the functional annotation of the phosphorylation events from PhosphoSitePlus database. If the phosphorylation at a specific residue is reported to play a positive role in tumorigenesis, $C_i$ is +1; if a negative role, $C_i$ is −1; phosphosites with conflict functional annotations in the database were not considered in the analysis. Bootstrap was performed with 10,000 replications to determine statistical significance[69]: 22 PI3K-AKT pathway $F_i$ values were simulated by drawing from the $F_i$ values of all quantified phosphorylation events, $C_i$ annotations were fed to each of these simulated data points to calculate a (P). This process was repeated 10,000 times. Finally, *p* value were calculated as the sum(a(P) > 1.45) /10,000.

*Establishment of HGG tumorspheres from mouse HGG tissue*: Tumors from *TPM3-NTRK1* implantations were dissected from the brain, mechanically dissociated, filtered through a 40 μm filter (Fisher) and seeded in ultra-low attachment flasks (Corning). Cells were grown in Neurobasal Medium (Gibco) supplemented with B-27 (Gibco #12587010), N2 (Gibco #17502048), GlutaMax

(Gibco #35050061), Heparin (2 µg/mL, Stem Cell Technologies #07980) human FGF (20 ng/mL, Miltenyi Biotec #130093842) and human EGF (20 ng/mL, Miltenyi Biotec #130097751). Cells were maintained as tumorspheres and grown at 37 °C, 5% $O_2$, 5% $CO_2$. Tumorspheres were passaged using accutase dissociation.

**CRISPR library construction and screening**. A set of 90 gRNA oligos that target on 15 master regulator genes identified through the multiomics integrative analyses and additional 6 nontargeting control gRNAs with no detectable match to mouse genome were designed for array-based oligonucleotide synthesis followed with library construction into pDUAL-vector-GFP[52]. Two individual gRNAs targeting on each gene was cloned into each pDUAL-vector-ZsG and three different vectors were designed for each gene. Unique binding of each gRNA was verified by sequence blast in the whole mouse genome. Mini gRNA library was constructed at transOMIC technologies. The mouse HGG cells were overexpressed with lentiviral Cas9 followed with selection by blasticidin as described previously[70]. The Cas9-expressing stable cells were re-infected with lentiviral-gRNA-library at M.O.I. = 0.1. The gRNA sequences were recovered by genomic PCR and deep-sequencing. The gRNA sequences are described in Supplementary Data 4i. The raw FASTQ data were de-barcoded, mapped to the original reference gRNA library. Counts for each gRNA were extracted and used for differential presentation analysis.

**Reporting summary**. Further information on research design is available in the Nature Research Reporting Summary linked to this article.

## Data availability

Raw genomic and proteomic data that support the findings of this study have been deposited in GEO with an accession number: GSE114331, and ProteomeXchange with an accession number: PXD005360. Analyzed data are included in this published article and its supplementary files. Source data underlying Figs. 1b, d, 2a−e, 3b−f, 4a−d, 5b, c, 6a −d, 7a−f, and 8b are provided as a Source Data file.

## Code availability

Code in this study was deposited on GitHub (https://github.com/hongwang198745/HGG_Source_Code).

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

## Acknowledgements

This work was partially supported by NIH grants AG047928, AG053987, and GM114260 (J.P.); CA188516 and CA096832 (S.J.B.); and ALSAC (American Lebanese Syrian Associated Charities). The MS and RNAseq analyses were performed in the Center for Proteomics and Metabolomics and the Hartwell Center, respectively, and animal experiments were supported by the Center for In Vivo Imaging and Therapeutics at St. Jude Children's Research Hospital, partially supported by NIH Cancer Center Support Grant (P30CA021765). We thank Chunxu Qu for the helpful discussions and comments.

## Author contributions

J.P., S.J.B. and H.W. designed the research; H.W. performed the proteomics experiments; A.K.D., B.S.P. and L.D.H. provided mouse models and RNAseq and connections to human HGG data; H.W., T.I.S., J.-H.C., Y.L., X.W., S.Z. and M.N. implemented the pipeline for computational analysis; H.W., H.T., Z.W., M.N., B.B., V.P. and A.A.H. implemented the pipeline for proteomics analysis; A.K.D. and J.S. performed immuno-blot and IHC experiments; H.W. designed and performed the CRISPR-Cas9 screening assay, H.W., Y.Z. and C.L. analyzed the CRISPR-Cas9 data, H.S.T., A. Shir-inifard, S.T. and A. Sablauer quantified Ki67 IHC, and H.W., S.J.B. and J.P. prepared the manuscript.

## Additional information

**Competing interests:** The authors declare no competing interests.

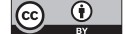

