## [Peer Review File · Nature Communications]

Reviewers' comments:

Reviewer #1 (HGG and mouse model expert) (Remarks to the Author):

This manuscript provides a general bioinformatics pipeline for identifying core signaling and master regulators in cancer proteomics. The authors take a few huge datasets from normal mouse brain, as well as from mouse models for two different high grade gliomas, dissecting whole proteome, phosphoproteome and transcriptomes, then describe systems biology approaches to identify functional modules master regulators, as well as using the resulting data, in comparison with human HGG transcriptome data to identify shared regulators. The methodology development here is impressive, and would be useful to researchers trying to work through similar large datasets. While no ground-breaking insights were made, the conclusions done through bioinformatic analyses were validated in biological systems.

The take home points, that NTRK mutation was a stronger activator of PDGFRA and of AKT than PDGFRA mutation, is certainly interesting, and unexpected. Can this result (7 b and c) be validated by immunoblotting of mouse HGG tumors, and by transfection of NTRK and PDGFRA mutation into indicator cells? Can the mechanism in Fig 7e be further delineated in cell-based systems as well?

Westerns in Fig 7 not displayed for me.

The authors validate the importance of EPHA2 and CD74 by citing papers in which these are already known to play roles in brain tumors, somewhat undermining the utility of their methodology to identify new insights. Can the authors analyze any of the more novel drivers identified to validate roles for these, and for CD74 in pediatric HGG?

Reviewer #2 (HGG expert) (Remarks to the Author):

The manuscript entitled: "Deep Multi-omics of Brain Tumors Identifies Signaling networks downstream of cancer driver genes" Wang and colleagues have modified murine cells representing two major mutation classes represented by high grade gliomas (HGGs). Major modifications (which the authors induced) include PDGFRA (D842V) and TPM3-NTRK1. Induced cells are then orthotopically injected into mice brain to create allograft models of HGGs. Tumor specimen are procured and analyzed deep-omics as the authors refer to their method. The strength of the manuscript is the bioinformatics analysis and correlation of data across platforms (RNA, protein, phosphor protein, pathway analysis). However, the impact of the presented work is reduced based on the following critiques:

Major comments:

1. The manuscript is mainly based on murine models generated by inducing murine astrocytes to express mutations in genes encoding PDGFRA and NTRKs. The authors reasons for not using human specimen is the presence of tumor heterogeneity and thus variation in data analysis. I find it difficult to follow this rational for two reasons: a) clinically, if we are to treat cancers, we will need to understand their true nature (heterogeneous in this case). Artificially simplifying cancer biology will not help developing effective treatments. b)if 'contamination' by heterogeneous cells is an issue, how are the authors accounting for contamination for healthy normal mouse brain cells that are being analyzed.
2. Since the majority of the data depends on the mouse model, a thorough characterization of the mouse model (histologically at least) is warranted. Is tumor a uniform mass? Is it infiltrating etc?
3. The cells that are intracranially injected are induced to express mutant genes! How many copies of each mutant expressing plasmid is induced? Is this taken into account as far as protein expression and true representation of human tumors?
4. The major distinction is that mutation in these genes do not constitute the major driver mutations for HGGs. The claim that this data represent true HGG biology should be toned down.
5. The manuscript would truly benefit of a more thorough validation across human specimen either

histologically, or by western etc.

6. Figure 3b, c, d, and e, are too general and not informative. What is the significant of these generic heatmaps?

7. Figure legends are not clear and do not reflect the generated data. For example, what are some of the elements indicated in Figure 3f? What is log₁₀FDR representing? False discovery rate? And if so, why is the higher FDR the better (as it seems to be the case according to the heatmap).

Minor comments:

1. Sentence "...and some well known..." line 93 is vague

2. Statement "Deep proteomics..." line 109 is vague and confusing

3. Summary sentence (line 117) should be adjusted to reflect the findings of this manuscript... is it truly the deepest analysis?

4. Claim "collectively, our comprehensive kinase..." is too strong. I believe this manuscript may represent a subtype of HGGs that have the two studied mutations and NOT all HGG tumors.

Reviewer #3 ((phosphoproteomics, proteomics, genomics and cancer expert) (Remarks to the Author):

The authors of this manuscript have used system biology approaches to study the proteome, phosphoproteome and transcriptome of HGG mouse models driven by mutant PDGFRA (D842V) or TPM3-NTRK1 fusion proteins. The mass spectrometry methods and data analysis pipelines used by the authors have generated deep HGG proteomics datasets and identified activation of multiple oncogenic pathways including PI3K-AKT signaling pathway. The manuscript is well-written and brings new insights to understanding the mechanisms of HGG development and progression. However, there are a few concerns that should be addressed before the manuscript is acceptable for publication:

Major:

1. In this study, authors used p53^{-/-} astrocytes with overexpression of oncogenic PDGFRA (D842V) or TPM-NTRK1. Interestingly, TPM-NTRK1 overexpression can dramatically elevate PDGFRA expression which is even higher than cells with overexpression of PDGFRA (D842V) (figure 7a-b). However, in figure 1d, the authors showed that PDGFRA expression level is much higher in PDGFRA cells than in TPM-NTRK1 cells. The authors should address this discrepancy and provide peptide sequences used for the differential quantification.

2. In addition to PDGFRA, the authors have demonstrated that several other RTKs were also upregulated in TPM-NTRK1 cells. This claim would be much more reinforced if the authors could show that HGG cell lines or clinical samples with endogenous TPM-NTRK1 mutations also have higher RTK expression than normal or tumors/cell lines with PDGFRA (D842V) mutation. Moreover, PDGFRA upregulation is one the key discoveries in this study. The authors should include the PDGFRA results in figure 8b and 8c-d panels.

3. The authors have shown that mice xenografted with TPM-NTRK1 cells had significantly shorter survival times than mice with PDGFRA cells. Is this the same in human HGG? If yes, this should be included in the discussion.

4. Activation of PI3K-AKT signaling pathway is a major discovery of the study. The authors could provide western blot evidence to confirm that AKT and downstream signaling molecules are indeed activated.

5. The authors should provide information on whether the protein FDR and report ion interference/isolation interference were used for data analysis.

Minor:

1. Figure 3d-e: The authors should provide the color keys and more detailed description for sizes of the circles/nodes, as well as the edges.

2. The resolution for figure 4b-c and figure S5a-b are too low. It is hard to tell some of the protein names. This should be fixed.
3. In lines 108 and 110, the authors should provide more information to define what 84% is referring to?
4. It is not clear what the three clusters represent (in Figure 4a and line 165).

Reviewer #4 (cancer systems biology and machine learning expert) (Remarks to the Author):

Overall Evaluation

The authors provide a very interesting data set regarding mouse models of high-grade glioma (HGG) driven by two different transgenic receptor tyrosine kinase oncogenes. They present a high-quality, high-coverage proteomic, phosphoproteomic, and transcriptomic data set together with standard analyses such as pathway-, kinase-, and transcription factor activity scoring and try to link the results of these. Their analyses point to AKT1 activity and an associated feedback loop to other RTKs as main determinant for oncogenic potency of HGG. AKT1 seems to be more active after NTRK1-induced oncogenesis than after PDGFRA, supported by the phenotypic response.

However, the manuscripts more or less stop at the point where they could potentially uncover very interesting biological insights, in particular on what exactly determines the observed differences in the phosphoproteomic responses to the oncogenes NTRK1 and PDGFRA, since the general patterns of proteomic/phosphoproteomic/transcriptomic responses seem to be more or less similar, albeit not in strength.

Major Comments

1 The bioinformatics analysis. What the authors call "a novel bioinformatics pipeline," feels a mixed set of arbitrarily chosen standard analyses. Different approaches come throughout the manuscript, and different choices are made, without explanation, and information is often lacking. Just a few examples:

1.1. In line 195: "TF activities were derived from target gene expression in either transcriptome or clustered proteome (WP-Cs), resulting in two lists of 47 and 46 TFs"

Why using proteome for TF activities, if TF what they do is to generate transcripts? proteome is generated from transcript, and therefore is affected also by pos-translational (and post-transcriptional) processes.

Furthermore, in line 262: "The TF activities were estimated by phosphorylation of active sites (Fig. 7e) and the target gene expression (Fig. 5)" Why now is phosphorylation used, and not proteomics as in line 195?

Also on TF scores, why ENCODE was used and not other resources?

- Lines 209 et seq. describe the reconstruction of a kinase-TF network for HGG. This step is insufficiently explained, also taking into account the methods part. Specifically how do the authors integrate 'consistent co-activation patterns' with the prior knowledge? Why do they get a comparably small network?

1.2. Kinase activities. Why using IKAP from multiple methods to compute kinase activities?

And the kinase-substrate information from PhosphoSitePlus was used. Did the authors use only the mouse-specific interactions for computation of the kinase activities or did they also map human

interactions to mouse proteins?

Overall, how big was the overlap between prior knowledge and the data, i.e. how big were the different substrate sets? See also Supplemental Figure 6, lines 62/63: Also done for other kinases?

1.3. The explanation of the pathway scoring method is not sufficient and should be expanded for better understanding. Additionally, how did they authors handle conflicts concerning Ci (the functional annotation of phosphosites)? How does their adapted method compare to other pathway scoring methods?

1.4. Authors should provide the code of their analysis for transparency and reproducibility

2. Differentially expressed phosphosites should be computed after correcting the phosphorylation data for the proteomic information since otherwise variations in the proteome will dominate the phosphosite information. If it was done, the text should be expanded to include this information, since neither from the main text nor the methods description, it becomes clear if the correction was performed.

3. As mentioned above, the finding that AKT1 is differentially activated after NTRK1 compared to PDGFRA feels underwhelming as the main result of the paper but should rather be the starting point to uncover what drives and determines this difference, e.g. by diving into the details of the phosphoprofiles of the proteins transmitting the signal from the oncogene to AKT1.

3.1. related to this, the authors found higher expression of RTKs after NTRK1-induced oncogenesis compared to PDGFRA. However, they did not comment on their signaling activity or their phosphorylation status, which could be considered more meaningful biologically and which should be already in the data they obtained.

4. the reference back to human data at the end of the manuscript feels unconnected to the rest and does not add essential results. Also the rationale behind the multi-species analysis and how it was exactly done is not clear.

Minor Comments

- In line 65 et seq., the authors point out that the analysis of patient samples is complicated by the "paradigmatic inter- and intratumoral heterogeneity" of human surgical HGG specimens. While they address the issue of intertumoral heterogeneity by using the same cell lines for the later experiments, this must not necessarily be true for the subject of intratumoral heterogeneity, which is not further addressed in the manuscript.

- Line 83: The authors should indicate already here that they used TMT labeling, not only in the methods part.

- Figure 1b is not discussed in the main text

- The information provided in lines 84 et seq. is a bit too technical for the main text and should be transferred into the methods part.

- Details about the PCA analysis are missing in the methods part (did they use all proteins/phosphosites or how did they handle NA values).

- Supplemental Figures 2 and 3 are mixed up. Furthermore, it should be indicated which correlation measure (Pearson, Spearman) was used.

- Figure 4b,c and Supplemental Figure 5 are of very low resolution.
- In all panels of Figure 5, the "Cortex" column in the heat-maps is useless and superfluous, since they show fold changes compared to the cortex condition. They should be removed.
- Figure 6d is more or less unnecessary, since the same is shown with Figure 6c and the numbers are mentioned in the text.
- Line 253: the results are phrased confusedly, comparing to Figure 1d. Maybe better: "showed higher PDGFRA wild type protein expression"
- Supplemental Figure 8 should show the data as fold changes or at least after log transformation
- What data does Figure 7c show? Gene expression or proteomic data?
- Figure 7d: why only EphA2? Did the authors did not try other RTKs or did it not work?
- Figures 7e and f can be condensed into one Figure for better visualization of the data. Furthermore, the discussion of these in the main text (lines 260 et seq.) should be expanded in order to better explain the approach. Why did the authors use MSigDB interactions and not -as previously- Encode? Why did they compute the TF activities again if they had them computed already in the previous section?
- Line 316: rate-limiting step?
- Line 321: mTOR was never mentioned in the manuscript before. The authors focused instead on AKT1.
- Line 424: What was the number of permutations? Did the authors check the p-values for convergence with the used number of permutations?

Reviewers' comments:

Reviewer #1 (Remarks to the Author):

This manuscript provides a general bioinformatics pipeline for identifying core signaling and master regulators in cancer proteomics. The authors take a few huge datasets from normal mouse brain, as well as from mouse models for two different high grade gliomas, dissecting whole proteome, phosphoproteome and transcriptomes, then describe systems biology approaches to identify functional modules master regulators, as well as using the resulting data, in comparison with human HGG transcriptome data to identify shared regulators. The methodology development here is impressive, and would be useful to researchers trying to work through similar large datasets. While no ground-breaking insights were made, the conclusions done through bioinformatic analyses were validated in biological systems.

1. "The take home points, that NTRK mutation was a stronger activator of PDGFRA and of AKT than PDGFRA mutation, is certainly interesting, and unexpected. a). Can this result (7 a and c) be validated by immunoblotting of mouse HGG tumors?"

As the reviewer pointed out, we validated the PDGFRA expression using immunoblotting of mouse HGG tumors (**Fig. 7b**). Moreover, the expression of selected Epha2 and its phosphorylation were validated by western blotting as well (**Fig. 7d**).

- b) And by transfection of NTRK and PDGFRA mutation into indicator cells?

The cell-based system may be simpler than mouse models to dissect disease mechanisms. Our mouse models are not generated by regular transgenic methods, but with intracranially transplanted astrocytes expressing NTRK or PDGFRA mutants, the mouse models are

essentially an extension of a cell-based system. So we did not transfect NTRK and PDGFRA mutation into another indicator cells.

c) Can the mechanism in Fig 7e be further delineated in cell-based systems as well?"

We performed a CRISPR-Cas9 screen with candidate master regulator transcription factors and showed that depleting Jun and Myc decreased growth of NTRK HGG cells.

2. "The authors validate the importance of EPHA2 and CD74 by citing papers in which these are already known to play roles in brain tumors, somewhat undermining the utility of their methodology to identify new insights. Can the authors analyze any of the more novel drivers identified to validate roles for these, and for CD74 in pediatric HGG?"

We agree with the reviewer's opinion. To validate the roles of novel master regulators identified through our multi-Omics integrative analysis, we spent about 1 year to establish cultures for primary *NTRK1* mutation-driven HGG cells and to perform a functional genetic CRISPR-Cas9 screening with a mini-gRNA library targeting for 9 master TFs and kinases, of which 5 master regulators are shown to be essential for the HGG viability, including master TFs (i.e. MYC and JUN) and key novel metabolic kinases (i.e. AMPK kinase subunits PRKAA1 and PRKAA2, and EEF2K), confirming the validity of the multi-Omics integrative approaches, and providing novel tumor vulnerabilities.

We added one full section for these new data in the text:

"CRISPR-Cas9 functional screening confirms the validity of the multi-Omics integrative approaches and identifies novel biological insights"

To examine if master regulators prioritized by the multi-Omics approaches are required for tumor survival, we first established the in vitro culture of HGG primary cells collected from NTRK-driven HGG mouse tumor tissues, and then designed a pooled mini-gRNA library for targeting these master regulators in a CRISPR-Cas9 analysis (Fig. 8a). We used a TransEDIT-dual CRISPR-Cas9 system⁵⁹, in which recombinant lentiviruses expressed dual gRNAs designed by a machine-learning approach to promote the functional ablation of genes (Fig. 8b). 6 non-targeting gRNAs were included as negative controls (Supplementary data 4j). Systematic experimental optimizations were performed (Supplementary Fig. 9), for example, stable expression of Cas9 in the HGG cells was confirmed by immunoblotting (Fig. 8c); gRNA integration was validated by fluorescence detection of ZsG (Fig. 8d); relatively even distribution of each gRNA in the pooled library was confirmed by deep-sequencing before screening (Fig. 8e), and screening was performed in triplicate for reproducibility.

We targeted two types of master regulators (kinases and TFs) during the CRISPR-Cas9 screening, including 9 genes derived from transcriptome and proteome data (Fig. 8f), and we also targeted 6 genes identified by cross-species comparisons with mouse and human tumors, each with 6 different gRNAs (i.e. 3 different dual-gRNA constructs). Dropout analysis was conducted to identify the essential regulators responsible for tumor survival. If 2 out of the 3 dual-

gRNA counts were significantly decreased after selection for 15 days compared to those in starting populations, the targeted gene was regarded to be important for tumor viability. Under this cutoff, none of the negative control gRNAs were enriched. On the other hand, 56% (5 out of 9) of the prioritized master regulators were shown to be critical for the HGG tumor growth. Strikingly, all three kinases (i.e. PRKAA1, PRKAA2, and EEF2K) regulating cell metabolism were found to contribute to HGG cell viability, providing a novel tumor vulnerability. Moreover, two TFs (Jun and Myc) were demonstrated to be positive hits in the screening. Given that RTK-PI3K-AKT induces a broad spectrum of downstream changes, pinpointing out Jun and Myc leads to valuable insights on how RTK fusions induce HGG tumorigenesis. Thus, this CRISPR-Cas9 validation screening unveils a novel tumor vulnerability of energy metabolism, and the involvement of Jun and Myc in NTRK fusion induced tumorigenesis, together confirming the validity of the multi-Omics integrative approach to discover master regulators.”

Figure 8

a

b

c

d

ZsG fluorescence after gRNA lentivirus transduction

e

f

Supp. Fig. 9

a

Quality controls and experimental optimization for transEDIT-dual CRISPR-Cas9 genomic screening

b

Primary HGG cells show mCherry fluorescence which co-express with *TPM3-NTRK1*

c

PCR products show expected amplicon size

d

Genomic functional screening using customized mini-gRNA library targeting cross-species conserved regulators

In addition, we also performed CRISPR-Cas9 screening to target on 6 genes derived from mouse-human cross-species omics integration, of which CD93 was found to be critical for the growth of *NTRK*-driven HGG primary cells. One of the reasons regarding the rest of candidates that failed to be enriched from the screening would be the low cutting efficiency of all gRNAs against the same gene. We also discussed these results and current limitations of our cross-species omics integration as follows:

“CRISPR-Cas9 screening was also designed to target 6 genes (Cd93, Cd74, Epha2, Spry1, Arhgap18, and Dab2) prioritized through the cross-species omics data integration, and showed that Cd93 was critical for the growth of NTRK-driven HGG primary cells (Supplementary Fig. 9d). Other candidates may have failed to be enriched from the screening because of low cutting efficiency of all gRNAs against the same gene. However, there were several limitations for our cross-species integration: (i) highly limited patient sample availability (i.e. only 8 patients with PDGFRA mutations and 3 patients with NTRK1 fusions) restricted the statistical power; (ii) large variations in the patients (e.g. tumor cells of origin, tumor growth environment, patient age and tumor grade) also confounded the data integration. Nevertheless, this approach would be useful for omics studies with large numbers of human samples.”

Reviewer #2 (Remarks to the Author):

“The manuscript entitled: “Deep Multi-omics of Brain Tumors Identifies Signaling networks downstream of cancer driver genes” Wang and colleagues have modified murine cells representing two major mutation classes represented by high grade gliomas (HGGs). Major modifications (which the authors induced) include PDGFRA (D842V) and TPM3-NTRK1. Induced cells are then orthotopically injected into mice brain to create allograft models of HGGs. Tumor specimen are procured and analyzed deep-omics as the authors refer to their method. The strength of the manuscript is the bioinformatics analysis and correlation of data across platforms (RNA, protein, phosphor protein, pathway analysis). However, the impact of the presented work is reduced based on the following critiques: “

Major comments:

1. “The manuscript is mainly based on murine models generated by inducing murine astrocytes to express mutations in genes encoding PDGFRA and NTRKs. The authors reasons for not using human specimen is the presence of tumor heterogeneity and thus variation in data analysis. I find it difficult to follow this rational for two reasons: a) clinically, if we are to treat cancers, we will need to understand their true nature (heterogeneous in this case). Artificially simplifying cancer biology will not help developing effective treatments.

We completely agree with the reviewer that understanding the heterogeneity of human HGGs is very important. Our study investigates the potential for an integrated proteomic and

phosphoproteomic approach to enhance our understanding of signaling within individual tumors. This is an important proof of principle for the use of such approaches for primary tumors. We clarified the rationale for the use of these models in the revised discussion on page 15-16:

“Recurrent mutations in the RTK/RAS/PI3K signaling axis occur frequently in virtually all adult glioblastomas, more than half of pediatric glioblastomas, and diverse other tumor types. While this implies that the PI3K pathway is an important therapeutic target, the response to small molecule inhibitors of the pathway is highly variable and often difficult to predict, likely due to varied consequences of specific mutations within the pathway, combinatorial effects with co-occurring mutations, complex feedback regulation within the pathway and cross-talk with other signaling pathways. Using model systems can help to clarify the contributions of specific mutations by eliminating some of the intertumoral heterogeneity caused by differing combinations of other mutations. In the present study, we investigated the sensitivity of integrated analysis of multiple-omics datasets to identify shared downstream pathways and differences in signaling in HGGs driven by two different glioma-associated RTK mutations in the same p53-null primary astrocyte population.”

Furthermore, use of the model system allowed us to use a CRISPR-Cas9 validation screening for 9 selected master TFs and kinases. 5 of these master regulators are shown to be crucial for the HGG viability, including master TFs (i.e. MYC and JUN) and key novel metabolic kinases (i.e. AMPK kinase subunits PRKAA1 and PRKAA2, and EEF2K), confirming the validity of the multi-Omics integrative approaches, and providing novel tumor vulnerabilities (**see more details in the pages 2-5 above**).

b) if ‘contamination’ by heterogeneous cells is an issue, how are the authors accounting for contamination for healthy normal mouse brain cells that are being analyzed.

We minimized the contribution of contaminating normal mouse cells by GFP-guided regional dissection as described in the methods on the revised manuscript page 19:

“Dissection of the focal regions of GFP-labeled HGG tumors was aided by visualizing with a fluorescence dissecting microscope to maximize tumor purity and minimize contaminating normal mouse cells. Dissected tumor tissue was snap frozen for proteome and transcriptome analyses.”

2. “Since the majority of the data depends on the mouse model, a thorough characterization of the mouse model (histologically at least) is warranted. Is tumor a uniform mass? Is it infiltrating etc?”

We added the following histological description on page 5 of the revised manuscript:

“Both models generated HGGs with highly mitotic pleomorphic tumor cells, many with features of astrocytic differentiation. The HGGs grew as focal masses with clear areas of invasion into the surrounding parenchyma at the boundaries of the tumor^{17, 21}.”

3. “The cells that are intracranially injected are induced to express mutant genes! How many copies of each mutant expressing plasmid is induced? Is this taken into account as far as protein expression and true representation of human tumors?”

We added a new Supplementary Fig. 3c and 3d (Inserted below) to investigate this question. Normalized RNAseq data was used to compare the expression of the human *NTRK1* (3c) or *PDGFRA* (3d) expression in mouse HGGs, control mouse cortex, or human HGGs from *Wu et al, Nat Genet, 2014*, categorized as HGGs carrying amplified wild-type *PDGFRA*, mutated *PDGFRA*, amplified and mutated *PDGFRA*, *TPM3-TRK1* fusion gene, fusion genes involving *NTRK2* or *3*, or HGGs without mutation in *PDGFRA* or *NTRK* genes. Expression of mouse orthologs for *NTRK1* (3c) or *PDGFRA* (3d) are shown in red. Analysis of variant allele frequency from whole genome sequencing data showed that human tumors with *TPM3-NTRK1* fusion gene carried the fusion gene as a subclonal population comprising only 9-20% of the tumor cells. Therefore, expression of the *TPM3-NTRK1* from bulk tumor RNA is a significant underestimate of expression of the *TPM3-NTRK1* fusion in human tumor cells. Mutant *PDGFRA* and *TPM3-NTRK1* in the mouse HGGs are expressed at levels that are relevant to the expression of these mutated genes in human HGGs.

4. “The major distinction is that mutation in these genes do not constitute the major driver mutations for HGGs. The claim that this data represent true HGG biology should be toned down.”

We apologize for the lack of clarity in our previous version. Activation of the RTK/Ras/PI3K pathway is a major driver, found in virtually all adult glioblastomas and the majority of pediatric HGGs. Many different mutations can activate this pathway in these tumors. As now stated in the revised results on pages 10:

“Collectively, our comprehensive kinase activity analysis enables the identification of master kinases and the downstream outcomes of kinase activation in two HGG tumor models in which PI3K pathway activation is driven by different receptor tyrosine kinase mutations that are found in human HGG.”

5. “The manuscript would truly benefit of a more thorough validation across human specimen either histologically, or by western etc.”

We agree with the review’s comments. But human HGGs with PDGFRA mutations or NTRK fusions were not available for biochemical analysis. Instead, we performed a CRISPR-Cas9 screening targeting on 6 genes derived from mouse-human cross-species omics integration. One gene (Cd93) was found to be critical for the growth of NTRK-driven HGG primary cells. We also discussed these results and current limitations of our cross-species omics integration (**also see details in page 5-6 above**).

6. “Figure 3b, c, d, and e, are too general and not informative. What is the significant of these generic heatmaps?”

We understand the concern of the reviewer that the heatmaps do not provide enough information on specific or individual proteins/phosphorylation. However, in the proteomics field, heatmaps are generally used to show the overall classification by global protein expression, representing a major step for quality control of proteomics datasets. For instance, the heatmaps were published in numerous high profile papers (*Zhang H. et al, Cell. 2016; Mertins P. et al, Nature. 2016; Zhang B. et al, Nature. 2014, Tan H. et al. Immunity. 2017; Stewart E. et al. Cancer Cell. 2018*).

7. “Figure legends are not clear and do not reflect the generated data. For example, what are some of the elements indicated in Figure 3f? What is log₁₀FDR representing? False discovery rate? And if so, why is the higher FDR the better (as it seems to be the case according to the heatmap).”

We re-wrote the figure legends with more detailed explanation to clarify our analysis.

FDR represents false discovery rate in the panel f. However, “-log₁₀ (FDR)” was the log transformation of the FDR, multiplied by “-1” to convert into a positive value. The higher the value of “-log₁₀ (FDR)”, the lower the FDR. Similar figures have been published in other recent studies (e.g. Stewart E. et al., *Cancer Cell*. 2018; Tan H. et al., *Immunity*. 2017).

Minor comments:

1. “Sentence “...and some well known...” line 93 is vague”

We re-wrote the sentence as “... phosphorylation events by Western blotting as reported in our previous study³¹”

2. “Statement “Deep proteomics...” line 109 is vague and confusing”

We removed the statement to avoid confusion.

3. “Summary sentence (line 117) should be adjusted to reflect the findings of this manuscript... is it truly the deepest analysis?”

It is the deepest analysis in a single-batch proteomics study in HGG to our knowledge, as we performed extensive fractionation and used long MS hours. To be conservative, we re-wrote the sentence as: “**one of the deepest HGG proteomic datasets**”.

4. “Claim “collectively, our comprehensive kinase...” is too strong. I believe this manuscript may represent a subtype of HGGs that have the two studied mutations and NOT all HGG tumors.”

We rewrote the sentence to specify the findings to the HGG subtypes we analyzed:

“Collectively, our comprehensive kinase activity analysis enables the identification of master kinases and the downstream outcomes of kinase activation in two HGG tumor models in which PI3K pathway activation is driven by different receptor tyrosine kinase mutations that are found in human HGG.” (Revised manuscript Page 10).

Reviewer #3 (Remarks to the Author):

“The authors of this manuscript have used system biology approaches to study the proteome, phosphoproteome and transcriptome of HGG mouse models driven by mutant PDGFRA (D842V) or TPM3-NTRK1 fusion proteins. The mass spectrometry methods and data analysis pipelines used by the authors have generated deep HGG proteomics datasets and identified activation of multiple oncogenic pathways including PI3K-AKT signaling pathway. The manuscript is well-written and brings new insights to understanding the mechanisms of HGG

development and progression. However, there are a few concerns that should be addressed before the manuscript is acceptable for publication.”

We appreciate this reviewer’s positive feedback.

Major:

1. “In this study, authors used p53^{-/-} astrocytes with overexpression of oncogenic PDGFRA (D842V) or TPM-NTRK1. Interestingly, TPM-NTRK1 overexpression can dramatically elevate PDGFRA expression which is even higher than cells with overexpression of PDGFRA (D842V) (figure 7a-b). However, in figure 1d, the authors showed that PDGFRA expression level is much higher in PDGFRA cells than in TPM-NTRK1 cells. The authors should address this discrepancy and provide peptide sequences used for the differential quantification.”

In the HGG models, we introduced a human version of the oncogenic *PDGFRA* (D842V) gene into mouse cells to generate HGG tumor. So we were able to quantify distinct peptides between human oncogenic PDGFRA and mouse wild type PDGFRA by MS. **Figure 1d** specifically quantified the mutated PDGFRA D842V, which we quantified by human-specific PDGFRA peptides, whereas **Figure 7a** was quantified by mouse-specific PDGFRA peptides to show that human NTRK fusion induced overexpression of mouse PDGFRA.

We re-wrote the sentences and figure legends to specify that quantifications were done differently based on human and mouse unique amino acid sequences by MS for clarification. Unique peptides used for these quantifications were highlighted below:

PDGFRA human

MGTSHPAFLVGLCLLTGLSLILCQLSLPSILPNEKVVQLNSSFSLRCFGESEVSWQYP 60
MSEESSDVEIRNEENNSGLFVTVLEVSSASAAHTGLYTCYNNHTQTEENELEGRHIYIY 120
VPDPDVAFVPLGMDTYLVIVEDDDSAIIPCRRTDTPETVTLHNSEGVPVASYDSRQGFNG 180
TFTVGPYICEATVKGK**KFQTI**PFNVYALKATSELNLEMDARQTVYKAGETIVVTCVAFVNN 240 197-223
EVDVLQWTPYGEVKGKGITMLEEIKVPSI**KL**VYTLTV**VP**KA**TV**KDSGEYECARQATKEVK 300 270-284
EM**K**V**T**ISVHEKGF**E**IE**K**PT**F**EQ**L**EAVNLHEVRE**F**VVEVQAYPT**P**RI**S**WLKDNLT**L**IENL 360 303-334
TEITTDVKEIQEIRYRSK**L**K**L**IRAK**E**EDSGHYT**I**VAQ**N**EDAVKSY**T**F**E**LL**T**Q**V**PS**I**L**D**I 420 385-435
VDD**H**H**G**S**T**G**G**Q**T**V**R**C**T**A**E**G**T**P**L**P**D**I**E**W**M**I**C**K**I**K**C**N**N**E**T**S**W**T**I**L**A**N**N**V**S**N**I**I**T**E**I**H**S**R**D** 480
RSTVEGRVTF**A**K**V**E**E**T**I**A**V**R**C**L**A****K**N**L**L**G**A**E**N**R**E**L****K**L**V**A**P**T**L**R**S**E**L**T**V**A**A**A**V**L**V**L**L**V**I**V**I** 540 492-501, 504-513, 515-524
SLIVLVV**I**W**K**Q**K**P**R**Y**E**I**R**W**R**V**I**E**S**I**S**P**D**G**H**E**Y**I**Y**V**D**P**M**Q**L**P**Y**D**S**R**W**E**F**P**R**D**G**L**V**L**G**R**V**L**G** 600 560-586
SG**A**F**G****K**V**V**E**G**T**A**Y**G**L**S**R**S**Q**P**V**M**K**V**A**V****K**M**L**K**P**T**A**R**S**E**K**Q**A**L**M**S**E**L**K**I**M**T**H**L**G**P**H**L**N**I**V**N**L** 660 606-618, 627-635, 638-646
LG**A**C**T**K**S**G**P**I**Y**I**T**E**Y**C**F**Y**G**D**L**V**N**Y**L**H**K**N**R**D**S**F**L**S**H**H**P**E**K**P**K****K**E**L**D**I****F**G**L**N**P**A**D**E**S**T**R**S**Y** 720 703-720
V**I**L**S**F**E**N**N**G**D**Y**M**D**M****K**Q**A**D**T**T**Q**Y**V**P**M**L**E**R**K**E**V**S**K**Y**S**D**I**Q**R**S**L**Y**D**R**P**A**S**Y**K**K**S**M**L**D**S**E**V**K**N** 780 735-754, 759-780
L**L**S**D**D**N**S**E**G**L**T**L**L**D**L**L**S**F**T**Y**Q**V**A**R**G**M**E**F**L**A**S**K**N**C**V**H**R**D**L**A**A**R**N**V**L**L**A**Q**G**K**I**V**K**I**C**D**F**L**A 840 804-813, 822-831
R**V**I**M**H**D**S**N**Y**V**S**K**G**S**T**F**L**P**V**K**W**M**A**P**E**S**I**F**D**N**L**Y**T**T**L**S**D**V**S**Y**G**I**L**L**W**E**I**F**S**L**G**T**P**P**P**G**M**M** 900 852-861
V**D**S**T**F**Y**N**K**I**K**S**G**Y**R**M**A**K**P**D**H**A**T**S**E**V**Y**E**I**M**V**Q**C**W**N**S**E**P**E****K**R**P**S**F**Y**H**L**S**E**I**V**E**N**L**L**P**G**O**Y**K**K 960 914-932, 939-960
S**Y**E**K**I**H**L**D**F**L**K**S**D**H**P**A**V**A****R**M**R**V**D**S**D**N**A**Y**I**G**V**T**Y**K**N**E**D**K**L**K**D**W**E**G**G**L**D**E**Q**R**L**S**A**D**S**G**Y**I**I** 1020 964-972, 979-995
P**L**P**D**I**D**P**V**P**E**E**E**D**L**G**K**R**N**R**H**S**S**Q**T**S**E**S**A**I**E**T**G**S**S**S**S**T**F**I**K**R**E**T**I**E**D**I**D**M**M**D**D**I**G**I**D**S 1080
SDLVDSFL 1089 (34%)

PDGFRA mouse

MGTSHQVFLVLSCLLTG**P**L**I**S**C**Q**L**L**L**PSILPNEKIVQLNSSFSLRCVGESEVSWQHP 60
MSEEDDPNVEIRSEENNSGLFVTVLEVVNASAAHTGWYTCYNNHTQTEDESEIEGRHIYIY 120
VPDPMAFVPLGMDTSLVIVEEDDSAIIPCR**T**TD**P**ET**Q**V**T**L**H**N**N**G**R**L**V**PASYDSRQGFNG 180 151-167
TFSVGPYICEATVKG**R**T**F****K**T**S**E**F**N**V**Y**A**L**K**A**T**S**E**L**N**L**E**M**D**A**R**Q**T**V**Y**K**A**G**E**T**I**V**V**T**C**V**A**F**V**N**N** 240 199-222
EVDVLQWTPYGEV**R**N**K**G**I**T**M**L**E**E**I****K****L****P**S**I****K**L**V**Y**T**L**T**V**P**K**A****T**V**K**D**S**G**E**Y**E**C**A**R**Q**A**T**K**E**V**K** 300 256-266, 270-280
EM**K****R**V**T**ISVHEKGF**E**IE**P**T**F**EQ**L**EAVNLHEVRE**F**VVEVQAYPT**P**RI**S**WLKDNLT**L**IENL 360 304-347
TEITTDVQKSQ**E**T**R**Y**Q**S**K**L**K**L**I****R**A**K**E**D**S**G**H**Y**T**I**VAQ**N**EDDVKSY**T**F**E**L**S**T**L**V**P**A**S**I**L**D**I** 420 383-420
VDD**H**H**G**S**G**G**Q**T**V**R**C**T**A**E**G**T**P**L**P**E**I**D**W**M**I**C**H**I**K**C**N**N**D**T**S**W**T**V**L**A**S**N**V**S**N**I**I**T**E**L**P**R**R**G 480 421-435
RSTVEGRV**S**F**A**K**V**E**E**T**I**A**V**R**C**L**A****K**N**N**L**S**V**V**A**R**E**L****K**L**V**A**P**T**L**R**S**E**L**T**V**A**A**A**V**L**V**L**L**V**I**V**I** 540 492-501, 515-523
SLIVLVV**I**W**K**Q**K**P**R**Y**E**I**R**W**R**V**I**E**S**I**S**P**D**G**H**E**Y**I**Y**V**D**P**M**Q**L**P**Y**D**S**R**W**E**F****R**D**G**L**V**L**G**R**I**L**G** 600 560-586, 590-600
SG**A**F**G****K**V**V**E**G**T**A**Y**G**L**S**R**S**Q**P**V**M**K**V**A**V****K**M**L**K**P**T**A**R**S**E**K**Q**A**L**M**S**E**L**K**I**M**T**H**L**G**P**H**L**N**I**V**N**L** 660 601-618, 627-635, 638-647
LG**A**C**T**K**S**G**P**I**Y**I**T**E**Y**C**F**Y**G**D**L**V**N**Y**L**H**K**N**R**D**S**F**M**S**Q**H**P**E**K**P**K****K**D**L**D**I****F**G**L**N**P**A**D**E**S**T**R**S**Y** 720 702-718
V**I**L**S**F**E**N**N**G**D**Y**M**D**M****K**Q**A**D**T**T**Q**Y**V**P**M**L**E**R**K**E**V**S**K**Y**S**D**I**Q**R**S**L**Y**D**R**P**A**S**Y**K**K**S**M**L**D**S**E**V**K**N** 780 735-754, 759-780
L**L**S**D**D**D**S**E**G**L**T**L**L**D**L**L**S**F**T**Y**Q**V**A**R**G**M**E**F**L**A**S**K**N**C**V**H**R**D**L**A**A**R**N**V**L**L**A**Q**G**K**I**V**K**I**C**D**F**L**A 840 781-813, 822-831
R**D**I**M**H**D**S**N**Y**V**S**K**G**S**T**F**L**P**V**K**W**M**A**P**E**S**I**F**D**N**L**Y**T**T**L**S**D**V**S**Y**G**I**L**L**W**E**I**F**S**L**G**T**P**P**P**G**M**M** 900 852-861
V**D**S**T**F**Y**N**K**I**K**S**G**Y**R**M**A**K**P**D**H**A**T**S**E**V**Y**E**I**M**V**Q**C**W**N**S**E**P**E****K**R**P**S**F**Y**H**L**S**E**I**V**E**N**L**L**P**G**O**Y**K**K 960 939-960
S**Y**E**K**I**H**L**D**F**L**K**S**D**H**P**A**V**A****R**M**R**V**D**S**D**N**A**Y**I**G**V**T**Y**K**N**E**D**K**L**K**D**W**E**G**G**L**D**E**Q**R**L**S**A**D**S**G**Y**I**I** 1020 964-972, 979-995
P**L**P**D**I**D**P**V**P**E**E**E**D**L**G**K**R**N**R**H**S**S**Q**T**S**E**S**A**I**E**T**G**S**S**S**S**T**F**I**K**R**E**T**I**E**D**I**D**M**M**D**D**I**G**I**D**S 1080
SDLVDSFL 1089 (38%)

Identified human PDGFRA sequence specific peptides: **NNNNNNNN**

Identified mouse PDGFRA sequence specific peptides: **NNNNNNNN**

Identified mouse and human PDGFRA shared sequence peptides: **NNNNNNNN**

2.1 “In addition to PDGFRA, the authors have demonstrated that several other RTKs were also upregulated in TPM-NTRK1 cells. This claim would be much more reinforced if the authors could show that HGG cell lines or clinical samples with endogenous TPM-NTRK1 mutations also have higher RTK expression than normal or tumors/cell lines with PDGFRA (D842V) mutation.”

We agree with the reviewer’s comment, and we actually did RNAseq analysis on the human specimen and analyzed the RTK expression in human specimen at transcripts level. Out of these 5 RTKs in figure 7, we only observed significant overexpression of Epha2 in patients with NTRK fusions compare to PDGFRA mutations (**Inserted figure**). The human sample analysis is complicated by intertumoral variation in the other mutations that are present in the tumor in addition to PDGFRA or NTRK mutation, and also by potential differences in the tumor cells of origin or tumor growth environment, both of which could be influenced by patient age and tumor location.

2.2 “Moreover, PDGFRA upregulation is one the key discoveries in this study. The authors should include the PDGFRA results in figure 8b and 8c-d panels.”

Due to the limitations of human tumor specimens (described above), we did not observe statistically significant upregulation of PDGFRA in these human samples.

3. “The authors have shown that mice xenografted with TPM-NTRK1 cells had significantly shorter survival times than mice with PDGFRA cells. Is this the same in human HGG? If yes, this should be included in the discussion.”

We examined the survival time of patient samples but did not identify statistical significance comparing HGGs with TPM-NTRK1 to HGGs without TPM-NTRK1. These fusion genes are enriched in HGGs from infants who have very low mutation burdens and longer survival compared to HGGs in older children. The relative contribution to survival time conferred by the NTRK fusion compared with the biological differences associated with tumor age and likely different cell of origin, as well as contributions of other mutations is not possible to assess with the small number of patient samples.

4. “Activation of PI3K-AKT signaling pathway is a major discovery of the study. The authors could provide western blot evidence to confirm that AKT and downstream signaling molecules are indeed activated.”

The activation of PI3K-AKT signaling pathway has already been examined in our previous publications, shown here:

a.

AKT pathway is active in PDGFRA D842V and other oncogenic PDGFRA mutations driven HGG tumors

a. western blot analysis of whole cell lysates from tissues of wild-type and mutant PDGFRA-driven brain tumors. Lysates from normal adult cortex (lanes N #1 and #2) were included as controls. Signaling pathway activation in PDGFRA-driven murine HGGs was assayed using the indicated antibodies. PDGFRA D842V tumor is highlighted with red arrow, Figure adapted from *Paugh B. Cancer Res. 2013*

b.

AKT pathway is active in NTRK fusions driven HGG tumors

b. Immunohistochemical analysis showed expression of FLAG-tagged NTRK fusion proteins, and elevated phospho-Akt in TPM3-NTRK1 and BTBD1-NTRK3 fusions driven tumor relative to surrounding normal tissue. Scale bar=50μm. Figure adpted from *Wu G. Nat. Genet. 2014*

Moreover, we spent substantial effort to validate other novel discoveries besides the PI3K-AKT pathway experimentally by establishing HGG primary cell cultures and CRISPR-Cas9 validation screening, confirming 5 master regulators to be crucial for the HGG viability, including master TFs (i.e. MYC and JUN) and key novel metabolic kinases (i.e. AMPK kinase subunits PRKAA1 and PRKAA2, and EEF2K) (see more details in the pages 2-5 above).

5. “The authors should provide information on whether the protein FDR and report ion interference/isolation interference were used for data analysis.”

We did use stringent FDR calculation method for proteins and the description of it has now been added in main text as well as the method section.

We understand that a caveat of the TMT method is that selected peptides are often contaminated by other co-eluting ions, leading to high noise signals to suppress quantitative ratios. The MS3 strategy has been developed to essentially eliminate this measurement inaccuracy, but requires more duty cycles with specific instruments with low resolution MS2 data for identification, which could influence peptide/protein identification. We recognized that the ratio suppression also affects experimental variations, and we did comparison of our MS2 based methods with MS3 based TMT methods analyzing the same sample, and demonstrated that quantitative ratio suppression has only a minor impact on statistical analysis (**Supplementary Figure 10, inserted below**). Moreover, the ratio suppression can be largely reduced by extensive sample fractionation, appropriate MS setting (e.g. narrow isolation window), and computer-assisted correction. To address the report ion interference issue in this study, we developed extensive fractionation by long gradient high resolution LC/LC-MS/MS to achieve deep proteome coverage and decrease ion suppression during quantification. In addition, we implemented y1-ion based noise detection and ratio correction to reduce ratio suppression (**Niu M. et al, Analytical Chemistry. 2017**).

Supp. Fig. 10

Fold change distribution and Z-value transformed distribution of MS2 and MS3 based peptide quantification comparing *NTRK*-driven HGG to Cortex using TMT labeling.

Minor:

1. “Figure 3d-e: The authors should provide the color keys and more detailed description for sizes of the circles/nodes, as well as the edges.”

We appreciate the reviewer’s comments and added more detailed description for the color, nodes and edges in the figure legend.

2. “The resolution for figure 4b-c and figure S5a-b are too low. It is hard to tell some of the protein names. This should be fixed.”

We provide high resolution images now.

3. “In lines 108 and 110, the authors should provide more information to define what 84% is referring to?”

We added more explanation for the 84% in line 108 for clarification and removed line 110 to avoid confusion. The sentence is now re-written as:

“We firstly applied a cutoff of FPKM >1 for the transcriptome to filter out low quality data. In 12,842 accepted transcripts, 10,838 (84%) corresponding proteins were mapped by MS.”

4. “It is not clear what the three clusters represent (in Figure 4a and line 165).”

More detailed description on what the three clusters represent has been specified:

“Hierarchical clustering analysis classified these kinase activities into multiple major clusters (Fig. 4a), resembling 3 major differential regulation patterns among cortex, PDGFRA, and NTRK HGGs in Figure 3c (i.e. PP-C1, PP-C2, PP-C5).”

Reviewer #4 (Remarks to the Author):

Overall Evaluation

“The authors provide a very interesting data set regarding mouse models of high-grade glioma (HGG) driven by two different transgenic receptor tyrosine kinase oncogenes. They present a high-quality, high-coverage proteomic, phosphoproteomic, and transcriptomic data set together with standard analyses such as pathway-, kinase-, and transcription factor activity scoring and try to link the results of these. Their analyses point to AKT1 activity and an associated feedback loop to other RTKs as main determinant for oncogenic potency of HGG. AKT1 seems to be more active after NTRK1-induced oncogenesis than after PDGFRA, supported by the phenotypic response.

However, the manuscripts more or less stop at the point where they could potentially uncover very interesting biological insights, in particular on what exactly determines the observed differences in the phosphoproteomic responses to the oncogenes NTRK1 and PDGFRA, since the general patterns of proteomic/phosphoproteomic/transcriptomic responses seem to be more or less similar, albeit not in strength.”

Major Comments

1 “The bioinformatics analysis. What the authors call “a novel bioinformatics pipeline,” feels a mixed set of arbitrarily chosen standard analyses. Different approaches come throughout the manuscript, and different choices are made, without explanation, and information is often lacking. Just a few examples:”

We agree that individual computational programs may not be novel, but it is the first time to integrate these programs for identifying potential cancer master regulators, using deep omics datasets. We now change “novel bioinformatics pipeline” into “integrated bioinformatics pipeline”.

1.1.1 “In line 195: “TF activities were derived from target gene expression in either transcriptome or clustered proteome (WP-Cs), resulting in two lists of 47 and 46 TFs” Why using proteome for TF activities, if TF what they do is to generate transcripts? proteome is generated from transcript, and therefore is affected also by pos-translational (and post-transcriptional) processes.”

We agree with the reviewer’s opinion on the biological gap between transcriptional regulation and protein level, thus we removed proteome data from TF activity analysis and re-analyzed the data. A new **Figure 5** and corresponding main text and method were also re-written in the revised manuscript.

1.1.2 “Furthermore, in line 262: “The TF activities were estimated by phosphorylation of active sites (Fig. 7e) and the target gene expression (Fig. 5)” Why now is phosphorylation used, and not proteomics as in line 195?”

We now re-visit the analysis using both whole proteome and phosphoproteome datasets for the TF analysis. The pipeline (in Fig. 7) is similar to that used in Fig. 5. Now Fig. 7 is revised (see page 1 above).

1.1.3 “Also on TF scores, why ENCODE was used and not other resources?”

ENCODE database contains only experimentally validated results, and is therefore the most stringent database to our knowledge. We then used the ENCODE for TF scores.

1.1.4 “- Lines 209 et seq. describe the reconstruction of a kinase-TF network for HGG. This step is insufficiently explained, also taking into account the methods part. Specifically how do the authors integrate 'consistent co-activation patterns' with the prior knowledge? Why do they get a comparably small network?”

We added more details on the method section on how the analysis was performed:

“Finally, to construct a putative kinase-TF network in HGG, we incorporated the relationships of kinase-substrate and TF-target from PhosphoSitePlus and Encode databases, respectively, and manually accepted kinase-TF networks with consistent co-activation patterns across different samples. For instance, AKT1 kinase showed different activity levels in three samples in an order of NTRK > PDGFRA > Cortex. The AKT1 is known to phosphorylate Brca1 at S686 residue.

The phosphorylated level of S686 also followed an order of NTRK > PDGFRA > Cortex.

Furthermore, Brca1-dependent transcripts were also elevated in an order of NTRK > PDGFRA > Cortex. Thus, we accepted the AKT1-Brca1 network.”

We appreciate that the reviewer recognized the change of network size. We simply reduced the size by avoiding repetition with the previous figure, but it is not accurate. Now we re-organize the figure and explain in the figure legend

“Kinases that do not have direct connections with downstream TFs were not shown to avoid redundancy with the previous figure” (see left).

1.2. “Kinase activities. Why using IKAP from multiple methods to compute kinase activities? And the kinase-substrate information from PhosphoSitePlus was used. Did the authors use only the mouse-specific interactions for computation of the kinase activities or did they also map human interactions to mouse proteins?”

1.2.1 “Why IKAP?”

IKAP can model the relationship of multiple kinase acting on a single substrate, while other available methods (e.g. KSEA) only assume a one-to-one kinase-substrate relationship. Thus we selected IKAP for our kinase activity modeling, as it mimics the complex kinase-substrate relationship in biology.

1.2.2 “Why PhosphoSitePlus database?”

To our knowledge, PhosphoSitePlus is currently the most comprehensive, freely available phosphoproteome database with the highest coverage of kinase-to-substrate information.

1.2.3 “only the mouse-specific interactions for computation of the kinase activities or did they also map human interactions to mouse proteins?”

Yes, we aligned the conserved phosphosites in rat, mouse and human proteomes, and used all possible information for the analysis.

1.2.4 “Overall, how big was the overlap between prior knowledge and the data, i.e. how big were the different substrate sets? See also Supplemental Figure 6, lines 62/63: Also done for other kinases?”

We performed the same analysis in Supp. Fig. 6 for other kinases in lines 62/63 (Supp. Fig. 7) now. Results are summarized in below table, overall, the overlap between prior knowledge and the data ranges from 52.6% to 83.3%, with a mean value of 67.8%.

Kinase	% Overlap	% different
AKT1	69.4	30.6
PRKAA1	72.7	27.3
CDK5	55.0	45.0
MAPK3	62.5	37.5
ATR	75.0	25.0
ATM	71.4	28.6
PAK1	52.6	47.4
FYN	83.3	16.7

1.3. “The explanation of the pathway scoring method is not sufficient and should be expanded for better understanding. Additionally, how did they authors handle conflicts concerning Ci (the

functional annotation of phosphosites)? How does their adapted method compare to other pathway scoring methods?”

1. We added more detailed explanation of the pathway in the method section now as below.

“

$$a(P) = \sum_{i=1}^k C_i * F_i / \sqrt{k}$$

In which K is the number of proteins with different activity relative to normal cortex samples, only PI3K-AKT pathway proteins with annotated functional phosphosites changes were accepted; F_i is the averaged log₂ fold change of DE phosphosites in proteini; C_i is the functional annotation of the phosphorylation events from PhosphoSitePlus database. If the phosphorylation at a specific residue is reported to play a positive role in tumorigenesis, C_i is +1; if a negative role, C_i is -1, phosphosites with conflict functional annotations in the database were not considered in the analysis. Bootstrap was performed with 10,000 replications to determine statistical significance: 22 PI3K-AKT pathway F_i values were simulated by drawing from the F_i values of all quantified phosphorylation events, C_i annotations were feeded to each of these simulated data points to calculate a (P). This process was repeated 10,000 times. Finally P value were calculated as the sum(a(P) > 1.45) /10,000.”

2. We manually examined the annotations of C_i first and simply discard the C_is with conflicts of functions.

3. There is no available method designed for pathway scoring using only phosphorylation events with known functional annotations before our method, not to say, considering the directionality of phosphorylation (C_i). Most of other tools use protein or transcript level to analysis pathway activity, which fail to capture real cell biology because signaling transductions are mainly accomplished through protein phosphorylation rather than protein/transcript level change. Our method is clearly superior to these methods in this case because the protein/transcript level of the PI3K-AKT pathway components barely changed in our data set. Nevertheless, the power of our method may be limited in cases where the pathways are regulated by protein/transcript changes or in cases with low known phosphorylation events identified.

1.4. “Authors should provide the code of their analysis for transparency and reproducibility.”

The codes for proteomics and other analyses that we adapted from others are listed below:

<https://www.stjuderesearch.org/site/lab/peng>

<https://bioconductor.org/packages/release/bioc/html/limma.html>

<https://cran.r-project.org/web/packages/WGCNA/index.html>

<https://ieeexplore.ieee.org/document/1565762>

<https://omictools.com/ikap-tool>

2. “Differentially expressed phosphosites should be computed after correcting the phosphorylation data for the proteomic information since otherwise variations in the proteome will dominate the phosphosite information. If it was done, the text should be expanded to include this information, since neither from the main text nor the methods description, it becomes clear if the correction was performed.”

Yes, we performed the whole proteome data normalization for kinase activity analysis because the kinase activity should be a direct measure of the signal contributed by the phosphorylation status.

However, for the pathway enrichment, we did not normalize it to whole proteome because the protein quantity is also an integral part of the measured pathway activity.

Regardless, we did not find significant differences between the normalized vs un-normalized phosphorylation results as the correlation shown below:

We further added both normalized data and data without normalization in the **Supplementary data 2a, 2b** in the revised version for reference.

3.1 “As mentioned above, the finding that AKT1 is differentially activated after NTRK1 compared to PDGFRA feels underwhelming as the main result of the paper but should rather be the starting point to uncover what drives and determines this difference, e.g. by diving into the details of the phosphoprofiles of the proteins transmitting the signal from the oncogene to AKT1.”

We agree with the reviewer that we should go steps further to find new biological insights beyond RTK oncogene activating AKT, and we decided to explore the role of other novel master identified upon the induction of RTK oncogene through our multi-Omics integrative analysis. We spent about 1 year to establish cultures for primary *NTRK1* mutation-driven HGG cells and to perform a mini gRNA library CRISPR-Cas9 functional genetic screening for 9 master TFs and kinases (**Fig. 8, Supp. Fig. 9, see page 2-5 above**). 5 of these master regulators are shown to be crucial for the HGG tumor cell viability, including master TFs (i.e. MYC and JUN) and key novel metabolic kinases (i.e. AMPK kinase subunits PRKAA1 and PRKAA2, and EEF2K), confirming the validity of the multi-Omics integrative approaches, and providing novel tumor vulnerabilities.

We added one full section for these new data in the text:

“CRISPR-Cas9 functional screening confirms the validity of the multi-Omics integrative approaches and identifies novel biological insights

To examine if master regulators prioritized by the multi-Omics approaches are required for tumor survival, we first established the in vitro culture of HGG primary cells collected from NTRK-driven HGG mouse tumor tissues, and then designed a pooled mini-gRNA library for targeting these master regulators in a CRISPR-Cas9 analysis (Fig. 8a). We used a TransEDIT-dual CRISPR-Cas9 system⁵⁹, in which recombinant lentiviruses expressed dual gRNAs designed by a machine-learning approach to promote the functional ablation of genes (Fig. 8b). 6 non-targeting gRNAs were included as negative controls (Supplementary data 4j). Systematic experimental optimizations were performed (Supplementary Fig. 9), for example, stable expression of Cas9 in the HGG cells was confirmed by immunoblotting (Fig. 8c); gRNA integration was validated by fluorescence detection of ZsG (Fig. 8d); relatively even distribution of each gRNA in the pooled library was confirmed by deep-sequencing before screening (Fig. 8e), and screening was performed in triplicate for reproducibility.

We targeted two types of master regulators (kinases and TFs) during the CRISPR-Cas9 screening, including 9 genes derived from transcriptome and proteome data (Fig. 8f), and we also targeted 6 genes identified by cross-species comparisons with mouse and human tumors, each with 6 different gRNAs (i.e. 3 different dual-gRNA constructs). Dropout analysis was conducted to identify the essential regulators responsible for tumor survival. If 2 out the 3 dual-gRNA counts were significantly decreased after selection for 15 days compared to those in starting populations, the targeted gene was regarded to be important for tumor viability. Under this cutoff, none of the negative control gRNAs were enriched. On the other hand, 56% (5 out of 9) of the prioritized master regulators were shown to be critical for the HGG tumor growth. Strikingly, all three kinases (i.e. PRKAA1, PRKAA2, and EEF2K) regulating cell metabolism were found to contribute to HGG cell viability, providing a novel tumor vulnerability. Moreover, two TFs (Jun and Myc) were demonstrated to be positive hits in the screening. Given that RTK-PI3K-AKT induces a broad spectrum of downstream changes, pinpointing out Jun and Myc leads to valuable insights on how RTK fusions induce HGG tumorigenesis. Thus, this CRISPR-Cas9 screening unveils a novel tumor vulnerability of energy metabolism, and the involvement of Jun and Myc in

NTRK fusion induced tumorigenesis, together confirming the validity of the multi-Omics integrative approach to discover master regulators.”

3.2 “related to this, the authors found higher expression of RTKs after NTRK1-induced oncogenesis compared to PDGFRA. However, they did not comment on their signaling activity or their phosphorylation status, which could be considered more meaningful biologically and which should be already in the data they obtained.”

We evaluated the signaling activity of NTRK1-induced oncogenesis compared to PDGFRA. It is clear that *NTRK1* induced higher signaling activity compare to PDGFRA. Result is shown in Fig. 6b (See below):

We also evaluated the phosphorylation levels of RTKs, *NTRK* fusion induced higher phosphorylation of RTKs compare to *PDGFRA* mutation (See below: **Differential phosphorylation of RTKs comparing *NTRK* to *PDGFRA* HGGs**)

4. “the reference back to human data at the end of the manuscript feels unconnected to the rest and does not add essential results. Also the rationale behind the multi-species analysis and how it was exactly done is not clear.”

We agree with the reviewer that there is a disconnection at the end of the manuscript, thus decided to remove it from the end of the manuscript, instead we added it under the section “multiple-omics integration identifies master regulators (kinases and TFs)” as a sub-section. The corresponding main figure is also moved to the supplementary. We further discussed the rationales and limitations of it in the discussion section and added more details in the method section to clarify how it was done.

Minor Comments

1. “- In line 65 et seq., the authors point out that the analysis of patient samples is complicated by the “paradigmatic inter- and intratumoral heterogeneity” of human surgical HGG specimens. While they address the issue of intertumoral heterogeneity by using the same cell lines for the later experiments, this must not necessarily be true for the subject of intratumoral heterogeneity, which is not further addressed in the manuscript.”

We agree with the reviewer and we remove the sentence in the main text.

2. “- Line 83: The authors should indicate already here that they used TMT labeling, not only in the methods part.”

As the reviewer suggested, we edited the sentence in Line 83 as:

*“**Tandem Mass Tag (TMT) labelling** was used to enable massively parallel proteome and phosphoproteome quantification of ten samples (**Fig. 1c**)”.*

3. “- Figure 1b is not discussed in the main text”

Figure 1b was actually discussed in the main text as “**Fig. 1a, b, referred to as PDGFRA HGG and NTRK HGG, respectively**”. We re-wrote the text as “**Fig. 1a, 1b ...**” for clarification

4. “- The information provided in lines 84 et seq. is a bit too technical for the main text and should be transferred into the methods part.”

We removed the technical description in line 84 et seq as the reviewer suggested.

5. “- Details about the PCA analysis are missing in the methods part (did they use all proteins/phosphosites or how did they handle NA values).”

PCA analyses were performed on all quantified proteins/phosphosites, and missing values were filtered out during the proteins/phosphosites quantification analyses before PCA analysis, so was not used in PCA analysis.

We added more details in the method section for clarification as:

“All quantified proteins and phosphopeptides were applied for the analyses. Missing values were filtered out during protein/phosphopeptide quantification, thus were not considered in PCA and hierarchical clustering analyses”.

6. “- Supplemental Figures 2 and 3 are mixed up. Furthermore, it should be indicated which correlation measure (Pearson, Spearman) was used.”

We appreciate the reviewer’s comments, and supplemental Figures 2 and 3 are in the right order now.

We used Pearson correlation and it is now specified in the supplemental figure legend.

7. “- Figure 4b,c and Supplemental Figure 5 are of very low resolution.”

We provided high resolution images now.

8. “- In all panels of Figure 5, the "Cortex" column in the heat-maps is useless and superfluous, since they show fold changes compared to the cortex condition. They should be removed.”

We appreciate the reviewer’s comments, and removed the “Cortex” columns in Figure 5 as the reviewer suggested.

9. “- Figure 6d is more or less unnecessary, since the same is shown with Figure 6c and the numbers are mentioned in the text.”

Figure 6d is removed now.

10. “- Line 253: the results are phrased confusedly, comparing to Figure 1d. Maybe better: "showed higher PDGFRA wild type protein expression"

We appreciate the reviewer’s great suggestion and re-wrote the sentence as *"showed higher PDGFRA wild type protein expression"*

11. “- Supplemental Figure 8 should show the data as fold changes or at least after log transformation”

We agree with the reviewer, since the data is already in supplementary table 3 and in the main text, we realized it is redundant to show it in the supplemental figure thus removed it.

12. “- What data does Figure 7c show? Gene expression or proteomic data?”

Figure 7c shows proteomic data. We re-wrote the sentence to “*Many other RTKs (EphA2, EGFR, FLT4, PTK7 and ROR2) also showed **higher protein expression** in NTRK HGG than PDGFRA HGG (Fig. 7c).*”

12. “- Figure 7d: why only EphA2? Did the authors did not try other RTKs or did it not work?”

We only tested Epha2 as a representative because the quantitative accuracy of our proteomics pipeline has already been demonstrated in many figures and well established in numerous other publications, thus we are confident the MS quantification of the rest of RTKs should be accurate as well.

Western blotting has confirmed 5 MS quantifications in supplementary Figure 1, and two more quantifications on PDGFRA in Figure 7b, MYC protein expression and phosphorylation on Figure 7f in the manuscript. These results all confirm the quantitative accuracy of the proteomics data. Moreover, the reliability of our proteomics pipeline has already been proved in numerous projects: Yang X. *Nat. Neurosci.* 2018; Shi H. *Immunity.* 2018; Cheng Y. *Nat. Neurosci.* 2018; Stewart E. *Cancer Cell.* 2018; Wang Z. *Nature.* 2018; Du X. *Nature.* 2018; Tan H. *Immunity.* 2018; Gong J. *Cell.* 2016; Lee KH. *Cell.* 2016.

13. “- Furthermore, the discussion of these in the main text (lines 260 et seq.) should be expanded in order to better explain the approach. Why did the authors use MSigDB interactions and not -as previously- Encode? Why did they compute the TF activities again if they had them computed already in the previous section?”

We appreciate the reviewer’s comments, and re-performed the TF analysis use both Encode and MsigDB for clarification.

We did not compute the TF activity again here. The sentence was re-written for clarification.

“We searched ENCODE and MsigDB databases to identify TFs that regulate the RTK transcription and validated TF activities by their protein levels or phosphorylation states (Fig. 7e).”

14. “- Line 316: rate-limiting step?”

We removed this description to avoid confusion.

15. “- Line 321: mTOR was never mentioned in the manuscript before. The authors focused instead on AKT1.”

Our phosphorylation profiling failed to identify known mTOR active site, but we observed activation of other mTORC components and downstream targets (e.g. 4EBP1, PRAS40, S6K), suggesting it is active.

The reason we focus on AKT instead is because:

- a. AKT is the major hub of deregulated PI3K-AKT signaling in our datasets
- b. AKT is more often selected to represent the activation of PI3K-AKT pathway in glioma studies.

16. “- Line 424: What was the number of permutations? Did the authors check the p-values for convergence with the used number of permutations?”

We permuted the columns of the expression matrix for 1000 times and then used it as null distribution to estimate the ANOVA p values for each protein following the Storey’s procedure (*J.D. Storey et al. PNAS. 2003*).

To check the convergence of the p-values, we repeated the above procedure for 10 times, and estimated the standard deviation (SD) of p value estimation for each protein (**See Figure below**). The estimation is very accurate as shown by the SD distribution, with the median of

0.0017, and 95% quantile of 0.015, highlighting the accuracy and convergence of the estimation process.

We revised our method to better clarify the process:

“The analysis was performed by ANOVA-based comparison of cortex, NTRK HGG, and PDGFRA HGG with P values estimated by permutation (1000 times) following Storey’s procedure and then adjusted by the Benjamin Hochberg method.”

Standard deviation (SD) distribution of estimated p values by permutation. For each protein, SD is calculated by 10 times of p value estimation.

Reviewers' comments:

Reviewer #1 (Remarks to the Author):

Revised manuscript addresses issues raised in prior review

Reviewer #2 (Remarks to the Author):

The manuscript by Wang and colleagues provides multi-platform analysis of tumor specimen to map the cancer signaling pathways. The authors have incorporated major revisions in response to reviewers' critiques. As a result, the manuscript quality and clarity has enhanced.

However, there are few remaining issues that will require author's attention/response.

1. I am not sure how feasible is to dissect "the focal regions of GFP-labeled HGG tumors was aided by visualizing with a fluorescence dissecting microscope." I agree with reviewer 1 and that the authors must take into account the potential for tumor/normal contamination.

2. References are not updated in the revised manuscript

3. Figure 3b, c, and d could be moved to supplemental figures.

4. Please expand the text to include FDR assessments as you described to reviewer #2, comment 7.

5. For proteomics analysis, the authors state that: "we were able to quantify distinct peptides between human oncogenic PDGFRA and mouse wild type PDGFRA by MS. Figure 1d specifically quantified the mutated PDGFRA D842V, which we quantified by human-specific PDGFRA peptides, whereas Figure 7a was quantified by mouse-specific PDGFRA peptides to show that human NTRK fusion induced overexpression of mouse PDGFRA."

Human and mouse protein sequences are 92% homologous and thus It seems very challenging to distinguish between the tryptic peptides specific to mouse or human. The homology map presented by authors is not accurate. For example, only few peptides seem to be specific to mice OR human post tryptic digestion (e.g. RTTDPETQVTLHNGRL)

However, a simple alignment of two protein sequences shows that some of the highlighted peptides (Authors response to reviewer #3) are indeed 100% homologous between the two species (e.g. KGITMLEEIK and KVTISVHEK) post tryptic digestion. Thus, it is not clear how many of the uniquely human and mouse peptides were identified? Was mass spectral analysis validated (e.g. by manual inspection for at least a few of the peptides) to ensure that the assignment of peptide to human or mouse (by automatic/software) were valid (for example, how were the 100% homologous peptides were assigned? Were these taken into account for quantification?

6. A validation of endogenous RTK over expression in human samples with TPN-NTRK1 mutations would be very informative. This point was also raised by reviewer #3, but was not adequately addressed.

Reviewer #3 (Remarks to the Author):

The revised version of the manuscript is greatly improved. All of the concerns raised by me have been addressed. I believe that the manuscript, in its current form, is a significant contribution to the field.

Akhilesh Pandey

Reviewer #4 (Remarks to the Author):

Overall:

The authors provided a significantly improved manuscript and adequately addressed many of the points raised during the last round of revisions. Especially the experimental validation by their gRNA screening adds a new quality to the work.

Some improvements/questions remain:

Major point:

- The authors end their discussion with "this integrated bioinformatics pipeline provides a general platform...". To follow up on this claim of a "general platform", the authors should provide all their code (for example in a dedicated Github repository or on Bitbucket), not just the identifiers of the tools they used, otherwise this claim is not justified.

Furthermore, even if the authors remove the claim, the code must be provided for transparency and reproducibility of their work.

Minor points:

- Discussion is very much a summary and only in parts a placement of their results in the field. A more extensive relationship to existing work will be most welcome.

- The argument to use ENCODE "ENCODE database contains only experimentally validated results, and is therefore the most stringent database to our knowledge. We then used the ENCODE for TF scores." is not quite true. There are many resources. For example, ReMAP includes ENCODE and many others. There are also curated, such as TTRUST, and integration of resources, such as DOROTHEA and CHEA. At this point we would not ask to rerun the analyses, though.

- Generally, throughout the manuscript there remain a few issues the authors should try to address. These involve reporting exact P values instead of "P value < 0.01", referring to FDR-corrected P values as q-values instead of as "FDR" (for example, the legend in Figure 5b reads "-log10(FDR)" but should instead read "-log10(q-value)", or p 3 | 60 should read "...phosphosites at an FDR of 1%...") and lastly providing more verbose and detailed figure legends. (For example, the figure legend for Figure 2b lacks the information which clustering methods was used, which distance was used for clustering, how the data were normalized or transformed before the analysis, and what message the figure should actually show.)

- Some of the figure panels are basic quality control plots (2def, 8b-e) and are not necessarily required to be part of the main manuscript. Instead, these plots could go into the supplement, followed by some reorganization of the figures; that is, panels e and f in Figure 1 and a and b in Figure 2 could be grouped together.

p 22 | 468 should be that start of a new section, at least of a new paragraph

p 15 | 311-314, why did the authors include transcriptomic in their study if it is unnecessary? It would maybe be good to have another introduction into the discussion.

p 11 | 219-230, why did the authors change from 6 TFs to 5 TFs for which they investigated targets sites? Did they not find any genes downstream of CEBP? If so, they should at least mention that.

p 9 | 188, the wording of "34 AKT substrates (70% of DE substrates)" is ambiguous. Did the authors mean that 70 of AKT substrates were differentially regulated? Or that of these 34 substrates, 70% were differentially regulated? It somehow cannot be the case that 34 AKT substrates are 70% of the more than 6000 DE phosphosites.

p 22: it remains unclear how exactly the authors performed "whole protein normalization". The authors should provide more methodological details here.

Reviewers' comments:

Reviewer #1 (Remarks to the Author):

Revised manuscript addresses issues raised in prior review.

We thank the reviewer for the positive comments.

Reviewer #2 (Remarks to the Author):

The manuscript by Wang and colleagues provides multi-platform analysis of tumor specimen to map the cancer signaling pathways. The authors have incorporated major revisions in response to reviewers' critiques. As a result, the manuscript quality and clarity has enhanced.

However, there are few remaining issues that will require author's attention/response.

1. I am not sure how feasible is to dissect "the focal regions of GFP-labeled HGG tumors was aided by visualizing with a fluorescence dissecting microscope." I agree with reviewer 1 and that the authors must take into account the potential for tumor/normal contamination.

We agree that there will always be some degree of contamination of non-tumor cells in diffuse glioma samples from human patients, or from mouse models, because of the diffuse growth properties of this disease. We did not mean to imply that dissection with the aid of a fluorescence dissecting scope excludes normal cells, it simply helps to identify optimal regions of highest tumor cell concentration with greater clarity than by unaided visualization. We accounted for noise introduced by variable amounts of contaminating normal tissue between samples by analysis of multiple replicates from tumors established from NTRK fusion or PDGFRA activation. Using this approach, we were able to focus on reproducible differences easily distinguished by possible low-level noise that may be introduced by contaminating normal cells. As detailed in lines 100-106: "*Principal component analyses and hierarchical clustering analyses revealed that the two RTK oncogenes drive distinct proteome, phosphoproteome and transcriptome profiles (Figs. 2a-2d). In the MS analysis, the intra-group replicate samples showed minimal variations with low standard deviation, whereas the inter-group comparisons exhibited differences with a much larger standard deviation (Supplementary Fig.2a, 2b). For transcriptome profiling, RNAseq replicates from a second cohort of HGGs displayed high reproducibility of these HGG mouse models ($R^2 > 0.95$, Supplementary Fig.3a, 3b, Supplementary data 3).*"

To further clarify this point for the reader, we also added "*Reproducible intragroup proteome, phosphoproteome, and transcriptome signatures showed that noise introduced by low-level contamination of normal cells did not mask robust differential signatures.*" in lines 417-419.

2. References are not updated in the revised manuscript

We updated the references.

3. Figure 3b, c, and d could be moved to supplemental figures.

We respectfully disagree with the reviewer's opinion, because panels 3b-d (protein clustering and network analysis) show critical information in the bioinformatics pipeline, including the size of each cluster (i.e. the number of DE proteins), the DE expression patterns, and the magnitude of alterations, as well as the major networks changed in HGG. Thus *the clustering and network information* in panels 3b-d is essentially complementary to the remaining panels showing *pathway enrichment*. For example, WP-C1 clustered proteins have an expression pattern of NTRK > PDGFRA > Ctl in panel 3b, which provides important details to the corresponding enriched pathways (shown in the first row of panel 3f). The panels 3b-f represent each step shown in panel 3a, which would offer a clear step-wise data interpretation for readers to understand our data processing.

However, if the reviewer and editor still recommend the removal of these panels to supplemental figures, we will revise the manuscript accordingly.

4. Please expand the text to include FDR assessments as you described to reviewer #2, comment 7.

We expanded the FDR assessment as the reviewer suggested, and also incorporated reviewer #4's new comments regard the FDR on line 907-909.

5. For proteomics analysis, the authors state that: "we were able to quantify distinct peptides between human oncogenic PDGFRA and mouse wild type PDGFRA by MS. Figure 1d specifically quantified the mutated PDGFRA D842V, which we quantified by human-specific PDGFRA peptides, whereas Figure 7a was quantified by mouse-specific PDGFRA peptides to show that human NTRK fusion induced overexpression of mouse PDGFRA." Human and mouse protein sequences are 92% homologous and thus It seems very challenging to distinguish between the tryptic peptides specific to mouse or human. The homology map presented by authors is not accurate. For example, only few peptides seem to be specific to mice OR human post tryptic digestion (e.g. RTTDPETQVTLHNNGL).

However, a simple alignment of two protein sequences shows that some of the highlighted peptides (Authors response to reviewer #3) are indeed 100% homologous between the two species (e.g. KGITMLEEIK and KVTISVHEK) post tryptic digestion. Thus, it is not clear how many of the uniquely human and mouse peptides were identified? Was mas spectral analysis validated (e.g. by manual inspection for at least a few of the peptides) to ensure that the assignment of peptide to human or mouse (by automatic/software) were valid (for example, how were the 100% homologous peptides were assigned? Were these taken into account for quantification?

We thank the reviewer for pointing out an error in the previous homology map that was manually generated. We re-inspected all identified peptides that matched to human or mouse PDGFRA protein. Although the human and mouse PDGFRA are 92% homologous at the single amino acid level, the related tryptic peptides are much more diverse in mass. Indeed, among the 42 identified tryptic peptides, 10 are unique to human sequence, 11 are specific to mouse sequence, and 21 are shared by both (see inserted table). These species-specific peptides provide reliable quantification for human or mouse PDGFRA protein.

Human specific	Mouse Specific	Shared peptide
K.EEDSGHYTIVAQNEDAVK.S	R.TTDPETQVTLHNNGR.L	K.QADTTQYVPMLEK.K
K.GFIEIKPTFSQLEAVNLHEVK.H	K.ATSELNLEMDAR.Q	K.GITMLEEIK.L
R.MAKPDHATSEVYEIMVK.C	K.NLLSDDDSEGLTLLDLSFTYQVAR.G	R.SLYDRPASYK.K
K.ATSELDLEMEALK.T	R.ILGSGAFGK.V	R.NVLLAQGK.I
K.LVYTLTVPEATVK.D	K.DLDIFGLNPADESTR.S	R.GMEFLASK.N
K.NLLGAENR.E	K.LVYTLTVPK.A	R.VDSDNAYIGVTK.N
K.ELDIFGLNPADESTR.S	K.TSEFNVYALK.A	R.SLYDRPASYKK.K
K.FQTIPFNVYALK.A	K.GFVEIEPTFGQLEAVNLHEVR.E	K.QALMSELK.I
K.SYTFELLTQVPSSILDVDDHHGGSTGGQTVR.C	R.EFVVEVQAYPTR.I	R.VTISVHEK.G
K.KVTISVHEK.G	K.SYTFELSTLVPASILDVDDHHGGGGQTVR.C	K.VEETIAVR.C
	R.AKEEDSGHYTIIVQNEDDVK.S	R.DGLVLGR.I
		K.MLKPTAR.S
		K.GSTFLPVK.W
		K.VVEGTAYGLSR.S
		K.QADTTQYVPMLEKESK.Y
		K.SMLDSEVK.N
		R.MRVSDSNAYIGVTK.N
		K.RPSFYHLSEIVENLLPGQYK.K
		K.LVAPTLR.S
		R.VIESISPDGHEIYVDPMLPYDSR.W
		K.IHLDFLK.S

6. A validation of endogenous RTK over expression in human samples with TPN-NTRK1 mutations would be very informative. This point was also raised by reviewer #3, but was not adequately addressed.

We agree with the reviewer that validation of RTK over expression in human samples would be valuable. We attempted to validate this using available data, but due to the very limited sample size of human HGG with TPM-NTRK1 fusion genes, we showed statistical significance of increased EphA2 (see inserted figure), but did not find statistically significant increases in expression of other RTKs. We think that this is not surprising because the human tumors have a wide variety of other mutations that will also influence cellular signaling and expression patterns. Also, the developmental origin of the tumors will contribute strongly to expression signatures. This may be particularly true for the HGGs with NTRK fusions that are enriched in infant patients, for which very few datasets are available.

Nevertheless, we believe this question will be more effectively addressed in the future as ever-increasing next-generation sequencing data analysis of patient samples in clinical settings will add greater statistical power. Use of the model systems allows us to generate a controlled comparison of the contribution of the different RTKs in a much more uniform genetic background, and using the same pooled population of cells of origin.

Reviewer #3 (Remarks to the Author):

The revised version of the manuscript is greatly improved. All of the concerns raised by me have been addressed. I believe that the manuscript, in its current form, is a significant contribution to the field.

We thank the reviewer for the positive comments.

Reviewer #4 (Remarks to the Author)

Overall:

The authors provided a significantly improved manuscript and adequately addressed many of the points raised during the last round of revisions. Especially the experimental validation by their gRNA screening adds a new quality to the work.

We thank the reviewer for the positive comments.

Some improvements/questions remain:

Major point:

- The authors end their discussion with “this integrated bioinformatics pipeline provides a general platform...”. To follow up on this claim of a “general platform”, the authors should provide all their code (for example in a dedicated Github repository or on Bitbucket), not just the identifiers of the tools they used, otherwise this claim is not justified. Furthermore, even if the authors remove the claim, the code must be provided for transparency and reproducibility of their work.

We agree with the reviewer and upload all programs (see figure below) in GitHub repository: https://github.com/hongwang198745/HGG_Source_Code

Minor points:

- Discussion is very much a summary and only in parts a placement of their results in the field. A more extensive relationship to existing work will be most welcome.

We expanded discussion to place downstream targets into a broader biological context.

- The argument to use ENCODE “ENCODE database contains only experimentally validated results, and is therefore the most stringent database to our knowledge. We then used the ENCODE for TF scores.” is not quite true. There are many resources. For example, ReMAP includes ENCODE and many others. There are also curated, such as TTRUST, and integration of resources, such as DOROTHEA and CHEA. At this point we would not ask to rerun the analyses, though.

We thank the reviewer for the clarification, and we added one sentence in lines 538-539: “ENCODE database contains only experimentally validated results, and is therefore used in this study.” in the revised version for clarification.

- Generally, throughout the manuscript there remain a few issues the authors should try to address. These involve reporting exact P values instead of “P value < 0.01”,

Exact P values are now reported in the text or in the supplementary data.

referring to FDR-corrected P values as q-values instead of as “FDR” (for example, the legend in Figure 5b reads “-log₁₀(FDR)” but should instead read “-log₁₀(q-value)”, or p 3 | 60 should read “...phosphosites at an FDR of 1%...”)

We corrected the description as the reviewer suggested.

and lastly providing more verbose and detailed figure legends. (For example, the figure legend for Figure 2b lacks the information which clustering methods was used, which distance was used for clustering, how the data were normalized or transformed before the analysis, and what message the figure should actually show.)

More detailed information was added in figure legends as the reviewer suggested.

- Some of the figure panels are basic quality control plots (2def, 8b-e) and are not necessarily required to be part of the main manuscript. Instead, these plots could go into the supplement, followed by some reorganization of the figures; that is, panels e and f in Figure 1 and a and b in Figure 2 could be grouped together.

Figures are re-organized as the reviewer suggested

p 22 | 468 should be that start of a new section, at least of a new paragraph

We started a new paragraph at p 22 | 468

p 15 | 311-314, why did the authors include transcriptomic in their study if it is unnecessary? It would maybe be good to have another introduction into the discussion.

We did not claim that transcriptome analysis was unnecessary. Our original sentence was “*As mRNA level is often only moderately correlated with protein level, there is a need to profile both the transcriptome and proteome to obtain a full picture of gene expression in cancer biology*” in p 15 | 311-314.

To avoid confusion, we added more discussion regarding this as the reviewer suggested at line 312-315 as:

“Both transcriptomic and proteomic analyses play indispensable roles for understanding the underlying central regulatory mechanisms in cancer biology. As mRNA level is often only moderately correlated with protein level, there is a need to profile both transcriptome and proteome to obtain a full picture of gene expression in cancer biology”.

p 11 | 219-230, why did the authors change from 6 TFs to 5 TFs for which they investigated targets sites? Did they not find any genes downstream of CEBP? If so, they should at least mention that.

We thank the reviewer for pointing it out. We added CEBP target genes in the Figure 5c, and revised the main text in line 219-230

p 9 | 188, the wording of “34 AKT substrates (70% of DE substrates)” is ambiguous. Did the authors mean that 70 of AKT substrates were differentially regulated? Or that of these 34 substrates, 70% were differentially regulated? It somehow cannot be the case that 34 AKT substrates are 70% of the more than 6000 DE phosphosites.

We removed “(70% of DE substrates)” in the revised version to avoid confusion

p 22: it remains unclear how exactly the authors performed “whole protein normalization”. The authors should provide more methodological details here.

Below are the steps for whole protein normalization (see inserted figure) using the following terms and a previous reported method (*Anal. Chem.* 2017, 89 (5), 2956-2963).

“Absolute protein abundance” presented by its corresponding peptide TMT intensities
“Relative protein abundance” presented by its TMT intensities divided by the mean of control samples

“Absolute phosphoprotein abundance” presented by its corresponding phosphopeptide TMT intensities

“Relative phosphoprotein abundance” presented by its TMT intensities divided by the mean of control samples

Step 1: Absolute protein abundance in each sample was first normalized by the average of the control samples (i.e. normal cortex) to obtain relative protein abundance.

Step 2: Relative protein abundance in each sample was log₂ transformed.

Step 3: Step 1 and step 2 were repeated for the phosphoproteome data.

Step 4: The “whole protein normalization” was performed by subtracting log-scaled relative protein abundance from log-scaled relative phosphoprotein abundance.

REVIEWERS' COMMENTS:

Reviewer #2 (Remarks to the Author):

The revised version of the manuscript is greatly improved. All of the concerns raised by me have been addressed. I have no further comments.